# Efficacy of Short-Term High Dose Pulsed Dapsone Combination Therapy in the Treatment of Chronic Lyme Disease/Post-Treatment Lyme Disease Syndrome (PTLDS) and Associated Co-Infections: A Report of Three Cases and Literature Review

**DOI:** 10.3390/antibiotics11070912

**Published:** 2022-07-07

**Authors:** Richard I. Horowitz, Phyllis R. Freeman

**Affiliations:** Hudson Valley Healing Arts Center, Hyde Park, NY 12538, USA; freemanp63@gmail.com

**Keywords:** dapsone, disulfiram, chronic Lyme disease, Post-Treatment Lyme Disease Syndrome, *Babesia*, bartonella, persisters, biofilms

## Abstract

Lyme disease and associated co-infections are increasing worldwide and approximately 20% of individuals develop chronic Lyme disease (CLD)/Post-Treatment Lyme Disease Syndrome (PTLDS) despite early antibiotics. A seven- to eight-week protocol of double dose dapsone combination therapy (DDDCT) for CLD/PTLDS results in symptom remission in approximately 50% of patients for one year or longer, with published culture studies indicating higher doses of dapsone demonstrate efficacy against resistant biofilm forms of *Borrelia burgdorferi*. The purpose of this study was, therefore, to evaluate higher doses of dapsone in the treatment of resistant CLD/PTLDS and associated co-infections. A total of 25 patients with a history of Lyme and associated co-infections, most of whom had ongoing symptoms despite several courses of DDDCT, took one or more courses of high dose pulsed dapsone combination therapy (200 mg dapsone × 3–4 days and/or 200 mg BID × 4 days), depending on persistent symptoms. The majority of patients noticed sustained improvement in eight major Lyme symptoms, including fatigue, pain, headaches, neuropathy, insomnia, cognition, and sweating, where dapsone dosage, not just the treatment length, positively affected outcomes. High dose pulsed dapsone combination therapy may represent a novel therapeutic approach for the treatment of resistant CLD/PTLDS, and should be confirmed in randomized, controlled clinical trials.

## 1. Introduction

Lyme borreliosis is the most common vector-borne disease in Europe and North America [1] and affects approximately 476,000 Americans per year [2] with over 200,000 cases annually in Western Europe [3]. At least two million individuals in the United States have been reported to be suffering from Post-Treatment Lyme Disease Syndrome (PTLDS) [4] with the incidence of PTLDS in Europe varying widely, likely due to differences in the case definitions and study designs [5,6]. Since up to 20% of Lyme borreliosis cases go on to develop PTLDS, healthcare costs and utilization after treatment can be significant [7], creating a burden for both the patient and healthcare system [8]. Estimates for medical costs related to chronic Lyme disease and PTLDS vary between USD712M and 1.3B each year in the USA [7].

Despite the discovery of Lyme disease over 40 years ago [9], the etiology of symptoms in chronic Lyme disease (CLD)/Post-Treatment Lyme Disease (PTLDS) is still a highly controversial topic [10,11]. The most frequent scientific hypotheses to explain persistent symptoms in CLD/PTLDS include immune evasion of *Borrelia* [12] with persistence of borrelial infection [13,14,15] or antigenic debris [16], persistent tick-borne co-infections including *Babesia* [17] and/or *Bartonella* [18], ongoing inappropriate immune activation and inflammation, or some combination of these [19]. Horowitz has identified up to 16 reasons why patients with CLD/PTLDS may remain ill called MSIDS (Multiple Systemic Infectious Disease Syndrome) [20]. This precision medical perspective assists in the definition, diagnosis, and management of PTLDS/CLD, and includes overlapping causes of inflammation (i.e., other co-infections, leaky gut, food/environmental allergies, environmental toxins [heavy metals, mold], mast cell activation, mineral deficiencies, and/or sleep disorders) along with downstream effects of inflammation (autoimmunity, hormonal dysregulation, mitochondrial dysfunction, resistant pain disorders, neuropsychiatric symptoms, Postural Orthostatic Tachycardia Syndrome [POTS] and liver dysfunction) [21]. All of these factors can potentially play a role in driving underlying symptomatology in CLD/PTLDS [22].

Recent advances in medicine have furthered a deeper understanding of the biology of *Borrelia*, which may help to also explain persistent inflammation in CLD/PTLDS, especially the discovery of stationary, persister, and biofilm forms of *Borrelia burgdorferi* [23,24,25]. These stationary forms in biofilms have been reported to be resistant to standard antibiotics and a primary source of inflammation [26,27]. Biofilm may harbor a heterogeneous population of spirochetes and round body forms, and *Borrelia* sp. within biofilms has been demonstrated to be much more difficult to eliminate, potentially increasing antibiotic resistance up to 1000 times [24,28,29]. There are two repurposed sulfa drugs, dapsone and disulfiram, that have demonstrated efficacy against resistant biofilm forms of *Borrelia burgdorferi* in both culture and clinical studies [30,31,32,33]. A search of the NCI compound collection suggested that disulfiram could have excellent borreliacidal activity against both the log and stationary phase *B. burgdorferi* sensu stricto, helping to lower inflammation [33], and two retrospective clinical studies using disulfiram showed efficacy in a cohort of chronically ill Lyme patients [32,33]. In vitro studies confirmed that dapsone alone and in combination with other intracellular antibiotics also has a significant effect on the resistant stationary phase biofilm form of *Borrelia* [34], as do essential oils including oregano oil, clove, cinnamon, and herbal compounds including Biocidin and Stevia [35,36,37,38]. There were two retrospective clinical studies that were conducted with a total of 300 patients on dapsone combination therapy (DDS CT), using a tetracycline, rifampin, dapsone, and hydroxychloroquine, along with essential oils, Biocidin and/or Stevia. The results demonstrated that this protocol was effective in relieving or improving up to eight major Lyme symptoms in those who had failed prior traditional antibiotic therapy [18,30]. Many individuals, however, relapsed after stopping treatment with lower dose dapsone (25–100 mg/day). In a follow-up clinical study, a higher dose of dapsone (100 mg BID, i.e., DDDCT) × 7–8 weeks showed improved clinical efficacy [39]. A total of 39 of 40 patients (98%) showed improvement of their tick-borne symptoms, and no patients had a worsening of symptoms post-therapy. A total of 45% had a resolution of all active Lyme symptoms post-treatment for one year or longer if there was no evidence of active co-infections, especially *Babesia*, *Mycoplasma,* and *Bartonella*. A total of 7 out of 12 patients (58%) with an EM rash and history of PTLDS remained in remission for one year or longer, and the remainder, 42%, showed a moderate improvement in underlying symptoms (37%) above their baseline functioning. Higher dose dapsone combination therapy, therefore, demonstrated clinical superiority to lower dose regimens [39] and the results were consistent with culture studies showing greater efficacy of dapsone and intracellular antibiotics at higher millimolar concentrations [34].

The present study using short-term high dose pulsed dapsone combination therapy (HDDCT) was, therefore, designed to evaluate several essential questions regarding the efficacy of these newer biofilm/persister medications in the treatment of CLD/PTLDS. Would dapsone pulsed at higher strengths (400 mg/day) have greater clinical efficacy than lower dose dapsone (200 mg/day, i.e., DDDCT), building on results of prior studies? Would combining two persister drugs (dapsone and disulfiram), increase the clinical efficacy and decrease the potential side effects? Would the dose of disulfiram and the length of time on DSF prior to using it in combination therapy with dapsone and other intracellular antibiotics (HDDCT) affect the long-term clinical efficacy? Could short-term, higher dose pulse dapsone combination therapy lead to better long-term remission rates than those seen in DDDCT, and avoid the need for longer term antibiotic therapy? Would the number of cycles of DDDCT prior to HDDCT influence the long-term efficacy? Finally, since prior studies showed that active infection with *Babesia* and especially *Bartonella* interfered with long-term remission in those who took DDDCT, were higher pulsed doses of dapsone combined with multiple intracellular antibiotics and biofilm agents effective against resistant co-infections, leading to better long-term clinical outcomes?

All three patients that are described in the following case presentations had a history of Lyme disease and *Babesia* and/or *Bartonella*, and took between 1 and 3 pulses of HDDCT. They remained in remission for time periods ranging between 3-months and 1-year with no further Lyme and tick-borne symptoms after completing this new protocol. A retrospective chart review of an additional 22 patients demonstrated that, in total, 84% of patients improved their tick-borne symptoms post-HDDCT despite some having active co-infections (*Babesia*, *Bartonella*), with 32% remaining in remission for 3-months or longer.

## 2. Case Presentations

### 2.1. Case 2.1

This 51-year-old white female with a past medical history (PMH) significant for attention deficit/hyperactivity disorder (ADHD), hyperthyroidism, and mononucleosis (8th grade), was diagnosed with Lyme disease in 2012 after developing an *erythema migrans* (EM) rash, with a positive ELISA and *Borrelia*-specific bands on a Western blot (23 kDa, Osp C). She lived in Michigan during the summer, and ‘pulled ticks off the dogs all the time’. In August 2012, she developed migratory pain in her feet, hands, left hip, knees, elbows, and forearms along with dizziness, headaches, memory/concentration problems, insomnia, and mood swings. Underlying chronic fatigue which was present for years significantly worsened during this time, along with the onset of sore throats, fevers, night sweats, hair loss, and a 20 lb. weight gain. Her PCP treated her with Pen VK 500 mg BID for 2 1/2-months and she felt significantly better while she was on antibiotics, although significant fatigue, joint pain, and cognitive difficulties persisted. She was, therefore, given steroids with methylprednisolone (a Medrol dose pack) several times, which led to temporary improvement (adrenal function was reported as low), although she relapsed immediately once she was off all treatment with antibiotics and steroids. Medications during her initial visit included venlafaxine 75 mg/day along with bupropion 150 mg/day for depression, dexmethylphenidate (Focalin XR) 20 mg/day, and several nutritional supplements (Vitamin D, berberine, oregano oil, adaptogenic adrenal herbs, D-ribose, monolaurin).

Her family history was unknown since she was adopted. As a young girl she lived in the Netherlands (her stepfather was Dutch), but she had no known tick exposure while in Europe, and she denied travel to malarial endemic areas. A review of systems was positive for the above history as well as bladder difficulties post-partum (G1, P2 [twins]). A physical examination revealed a well-developed, well-nourished white female in no apparent distress. She was 5 foot 9 inches, 181.2 lbs., afebrile, respirations were 14 per minute, BP was 102/80 sitting, with a pulse rate of 76 BPM, which remained relatively stable at 106/88 standing with a pulse rate of 88 BPM. Repeat BP and pulse rates sitting and standing were suggestive, however, of moderate Postural Orthostatic Tachycardia Syndrome (POTS), with an initial sitting BP/pulse rate of 124/79, 78 BPM, decreasing to 107/90, with her pulse rate rising to 101 BPM (regular) at 5 min standing. The rest of the physical examination was unremarkable, with normal examination findings for the cardiovascular, pulmonary, gastrointestinal, and neurological systems.

A records review revealed a positive Lyme IgM ELISA at 2.16 (normal range < than 0.90), 23 kDa (Osp C) on a local Western Blot, old exposure to EBV and negative autoimmune markers. A thyroid ultrasound showed 5 mm cystic nodules that were unchanged from prior reports (she was off methimazole for hyperthyroidism at the time of the report with normal thyroid functions). A comprehensive tick-borne panel was sent out to local and specialty labs to confirm exposure to Lyme disease, as well as associated co-infections including *Babesia*, *Ehrlichia*, *Anaplasma*, *Bartonella*, *Mycoplasma*, *Chlamydia pneumoniae*, *Rickettsia rickettsii* (Rocky Mountain Spotted Fever), *Coxiella burnetti* (Q-fever), *Franciscella tularensis* (Tularemia), and *Brucella*, along with viral titers (West Nile, cytomegalovirus [CMV], herpesvirus 1 [HSV-1], herpesvirus 2 [HSV-2], and herpesvirus 6 [HHV-6]) with polymerase chain reactions (PCR’s) for herpes viruses, along with an evaluation of hormone levels (adrenal, thyroids, growth hormone, sex hormones), mineral levels, complement studies and immunoglobulin levels, inflammatory markers, food allergies, and exposure to heavy metals via a 6 h urine dimercaptosuccinic acid (DMSA) challenge. All polymerase chain reactions (PCRs) and antibody tests were negative for other tick-borne diseases (TBD’s), except for a tularemia titer for *Franciscella tularensis* that was reported as equivocal at 1:20 + (Quest Diagnostics, Chicago, Illinois) and *Babesia duncani* titers which were positive (WA1 IgG antibody) at 1:256 through Quest Diagnostics (Normal range < 1:256), providing a potential etiology for her significant night sweats; A repeat Lyme Western Blot showed exposure to multiple *Borrelia*-specific bands (31 kDa, Osp A; weakly positive 34, Osp B, 39, and 83/93 kDa bands), and *Babesia* titers for *Babesia microti* and a *Babesia* fluorescence in situ hybridization (FISH) test from (IgeneX Laboratory, Milpitas, California) were negative. Hormone panels (sex hormones, thyroid hormones) failed to reveal other etiologies for the night sweats. *Mycoplasma pneumoniae* and *Chlamydia pneumoniae* IgG antibodies were positive (old exposure), as were HSV-1 (2.98, Normal range 0–0.89) and HHV6 antibody levels, which were positive at 1:160. The iodine levels were low (49, normal range 52–109 mcg/L), and complement C4a levels were elevated (1019.9; normal range 0–650 ng/mL). Otherwise, all inflammatory markers (sedimentation rate, high sensitivity C Reactive Protein [CRP]), and autoimmune markers were negative, as was a vascular endothelial growth factor (VEGF), an indirect marker of exposure to *Bartonella*.

The patient was initially treated for Lyme and babesiosis with hydroxychloroquine 200 mg PO BID, cefdinir 300 mg PO BID, azithromycin 250 mg PO BID (pulsed 4 days in a row per week), and nystatin 500,000 units, two PO BID, along with atovaquone/proguanil tablets (Malarone) 100/250 mg, two PO BID, grapefruit seed extract, two PO BID, oregano oil capsules (60 mg BID), and triple probiotics (Theralac, Ultraflora DF, *saccharomyces boulardii*) BID. A strict sugar-free and yeast-free diet was advised for both weight-loss and to keep down the risk of *Candida* overgrowth (she had a history of occasional *Candida* infections in high school). She was switched to atovaquone (Mepron) one tsp PO BID along with clindamycin 300 mg, two PO BID, and azithromycin 250 mg PO BID for her babesiosis after drenching night sweats persisted on her prior regimen, eventually rotating in lumefantrine/artemether (Coartem) 4 PO BID for three days for resistant babesiosis. Her EKG was within normal limits (WNL) on Coartem without any evidence of heart block or QT prolongation. This regimen helped decrease her drenching night sweats (they were present, but less intense), and she felt up to 90% of normal functioning after 2-months of treatment. Antibiotics were, therefore, discontinued since most symptoms improved, although daily resistant joint pain (hands, hips) persisted, requiring restarting antibiotics. She resumed doxycycline 200 mg PO BID, hydroxychloroquine 200 mg PO BID, nystatin 500,00 units, two PO BID with atovaquone/proguanil tablets, 100/250 two PO BID, and cryptolepis 30 drops twice a day.

At a one-month follow-up, her fatigue and joint pain had improved, and 2-months later her sweats were gone with no further headaches, dizziness, neck pain, or sore throats. Her anxiety and depression had also improved, along with decreased muscle and joint pain. She still complained, however, of mild to moderate joint pain, which was controlled with hydroxychloroquine; fatigue, which was present for the past 30 years, but improved compared to several months prior; and some cognitive difficulties. Her memory/concentration was better, but decreased processing speed persisted. Since she felt 95% of her normal functioning, we stopped all antibiotics again and placed her on low-dose naltrexone (LDN) 3 mg HS to help decrease central nervous system (CNS) microglial activation, along with 1000 mcg/day of methylcobalamine. Although her B12 levels were normal, she felt better and less sluggish with methylation and B vitamin support. Herbal support for her Lyme disease (Beyond Balance MC-BB2, 10 drops once a day and MC-BAB, 1–2 drops twice a day) was also instituted, which helped stabilize symptoms off antibiotics. A low carbohydrate, hypoglycemic diet (40% protein, 30% healthy fats, 30% complex carbohydrates) with small, frequent meals was suggested for her chronic fatigue, sugar cravings, and increased weight. She frequently suffered with severe episodes of mid-day fatigue requiring a nap, suggestive of significant reactive hypoglycemia and blood sugar swings/and or adrenal dysfunction. The low carbohydrate diet was also prescribed to help with weight-loss, which was medically necessary since a CAT scan of her abdomen and pelvis revealed hepatic steatosis of the liver. The scan was initially performed for chronic constipation and intermittent abdominal pain, which improved over time with increased fluids, fiber, and extra magnesium.

By May of 2015, several years after her initial presentation, her symptoms began relapsing with low grade fevers, occasional night sweats several times a month, hot flashes, “air hunger”, and increased joint pain with fatigue. We rechecked her Lyme and tick-borne titers, along with a CBC, CMP, CRP, ESR, HHV6, EBV titers, and TSH. A repeat Lyme Western Blot revealed increased *Borrelia*-specific bands over time (34 kDa, Osp B) and a slight increase in tularemia titers (1:40 positive, normal range < 1:20). Adrenal testing via a DHEA/cortisol salivary test revealed Stage II adrenal fatigue with low DHEA (43.74, normal range 106–300 pg/mL) and low cortisol levels mid-day (1.78, normal range 2.1–15.7 nmol/L). Her presentation was suggestive of a relapse of her Lyme and babesiosis, with overlapping adrenal dysfunction, so she was started back on herbal adrenal support and a previously effective regimen, including doxycycline 200 mg PO BID, hydroxychloroquine 200 mg PO BID, nystatin 50,000 units BID, Malarone 100/250 mg, two PO BID, cryptolepis 30 drops twice a day, and MC Bab-2, 15 drops twice a day. She noticed a Herxheimer reaction the first three weeks back on doxycycline, and then her flu-like symptoms improved, with decreased night sweats, hot flashes, fevers, and air hunger. Her fatigue and joint pain were also better, but persisted, so rifampin 300 mg PO BID and dapsone 50 mg every other day were added to her regimen along with Leucovorin 25 mg/day and L-methyl folate, 15 mg/day. Folic acid supplementation was added to minimize dapsone-induced anemia. Sleep remained restless and an in-home sleep study was ordered to rule out restless leg syndrome (RLS), which was negative for obstructive sleep apnea (OSA) or RLS. Magnesium L-threonate 200 mg HS was added to her regimen, which improved her sleep patterns. By September 2015, she had been on two months of minocycline 50 mg PO BID (doxycycline was changed to lower dose minocycline to minimize the possibility of sunburn during summer months), rifampin 300 mg PO BID, dapsone 50 mg QOD, Plaquenil 200 mg PO BID, and Leucovorin 25 mg/day. She felt significantly better on dapsone combination therapy (at low dose), with a resolution of night sweats × 3-months, air hunger, headaches, and hip pain, although mild joint pain in the hands continued to persist, with cognitive difficulties and name recall issues. CBCs remained stable on dapsone, as there was only a mild decrease in her hemoglobin (Hb) and hematocrit (Hct) (13.3/4l.6 decreased to 12.6/39.2), so dapsone dosing was increased to 50 mg/day along with increased minocycline dosing at 100 mg PO BID for better CNS penetration. By December 2015, she was functioning at 95% normal, with no further *Babesia* symptoms (night sweats were gone × 6-months).

As of February 2016, after being on minocycline with low-dose dapsone (50 mg) for 7-months, she felt better than she had in years, with significant improvement in her joint pain, memory, and concentration problems, with no further return of her significant fatigue, sweats, air hunger, palpitations, or headaches. She felt close to 100% normal, and antibiotics were stopped. By September 2016, she had been off dapsone for 7-months without a relapse, and felt “awesome”, with no further return of joint pain. There was always some underlying fatigue which had been present her entire life, so she was given a trial 2-months of mitochondrial nutritional support with glycosylated phospholipids, co-Q10, Nicotinamide adenine dinucleotide (NADH), and acetyl-L-carnitine along with a ketogenic diet, while continuing to exercise on a regular basis. A stricter low-carb diet and mitochondrial support did mildly improve her underlying fatigue. She remained in good health until April 2018 when she had an emergency total abdominal hysterectomy (TAH) for a prolapsed uterus and rectocele. Several months post-op, without any evidence of further tick bites, symptoms began to relapse again with low grade fevers for several weeks, sore throats, moderate fatigue, mild joint pain in her hands, along with some mild memory and concentration problems. Her moods remained stable on venlafaxine 100 mg per day, but there was some increased anxiety over her son who became suicidal and required institutionalization. Repeat tick-borne titers did not show any significant increase in *Borrelia*-specific bands (Osp B, 34 kDa remained positive) and *Babesia* titers as well as a *Babesia* FISH test from IgeneX laboratories was negative. A *Stachybotrys* titer returned positive with a history of recent mold exposure, and she was started on a detoxification regimen with glutathione, binders, and far infrared saunas, with continued focus on a healthy exercise regime and a low carbohydrate/ketogenic diet. This helped her underlying symptoms. Her energy improved, joint pain, and night sweats resolved, and she lost 20 pounds with her diet and exercise regimen.

During a routine medical follow-up in March 2020, the patient reported a back injury at L5/S1 and began to notice relapsing arthritic symptoms in her shoulders, hips, and hands. The joint pain did not come and go and migrate, and she complained of new onset of pruritus on a regular basis. Her history was suggestive of mast cell activation with histamine release, and possible leaky gut with reactions to foods that were high in nightshades (tomatoes appeared to increase her pain). We, therefore, sent off an IgE and IgG food panel as well as mast cell markers (histamine, chromogranin A, tryptase), repeated tick-borne testing, and asked her to send off a breath test for leaky gut. She was started on fexofenadine (Allegra) 180 mg once a day along with an H2 blocker (famotidine 10 mg). Tick-borne testing showed a new increase in her 39 kDa band on her Western Blot (Quest laboratories, Chicago, Illinois), and an IgG4 food allergy panel showed some intolerance to wheat, eggs, and millet with multiple low grade food sensitivities. All other tests were negative. Since histamine blockade with changes in her diet helped her underlying symptoms, she continued to stay off antibiotics and remained stable, feeling close to 100 percent normal until August 2020, when an increase in stress contributed to a significant relapse of her underlying Lyme symptoms. Her husband informed her that he wanted a divorce after 23 years of marriage, and she suddenly began to complain of moderate fatigue; moderate to severe joint pain, especially in her neck and hands; increasing night sweats; insomnia; and anxiety. There was no worsening of adrenal function and she started back on cefdinir 300 mg PO BID, hydroxychloroquine 200 mg PO BID, Zithromax 250 mg BID (4 days in a row per week), with atovaquone/proguanil 100/250 mg 2 PO BID with triple biofilm agents (oregano oil, Biocidin, Stevia) and triple probiotics. Trazodone 50 mg HS was used PRN for sleep and she began seeing a therapist/psychiatrist, who increased her venlafaxine to 225 mg per day. Although there was some mild benefit with this protocol, she still complained of increasing night sweats, worsening arthritis in her feet and ankles, fatigue, and increased cognitive difficulties. Hormone testing ruled out menopause or hyperthyroidism.

The patient remained relatively stable until early 2021, on and off antibiotics, when a repeat IgG Lyme Immunoblot in March 2021 showed a new 23 kDa (OspC) band and weakly positive 93 kDa band (IgeneX), consistent with reactivation of *Borrelia burgdorferi*, along with a positive *Babesia* FISH test, consistent with an active *Babesia* infection. A *Bartonella* FISH test was negative along with a normal level of VEGF. She was, therefore, switched to minocycline 100 mg PO BID, hydroxychloroquine 200 mg PO BID, rifampin 300 mg PO BID, dapsone 25 mg PO BID, leucovorin 25 mg per day, l-methylfolate 15 mg per day, and Malarone 2 PO BID, as this regimen had been effective in the past. February 2021, the patient decided to do a higher dose dapsone combination therapy due to her frequent relapses over the years, and increased her dapsone to 100 mg per day. She stayed on this protocol for two months but was still functioning at 50% of normal due to drenching night sweats, arthritis, muscle pain, and cognitive impairment. In April 2021, Malarone was changed to atovaquone (Mepron) 750 mg BID for her positive *Babesia* FISH and severe night sweats, along with cryptolepis 30 drops 3×/day and artemisinin one capsule 3×/day. A CBC remained stable on dapsone (hemoglobin 12.3/hematocrit 36.5), with a normal CMP. She signed an informed consent regarding the side effects of high dose dapsone combination therapy, and instead of taking 200 mg of dapsone and working up the dose, then increasing to 200 mg BID × 4 days, she went straight to HDDCT (200 mg BID) with minocycline, rifampin, hydroxychloroquine, and nystatin from 100 mg of dapsone/day. Despite complaining of significant nausea and vomiting one time on the protocol, she finished the protocol April 2021. By May 2021, off of the antibiotics for one month, she was feeling close to 100% normal, with no further fatigue, arthritis, headaches, night sweats, insomnia, or cognitive difficulties. Her baseline functioning was the best it had ever been, including feeling emotionally well and calmer. She had developed a mild yeast infection at the end of antibiotic therapy with some dietary indiscretions and took fluconazole 200 mg per day for 10 days post-therapy, while remaining on higher dose leucovorin and l-methylfolate during the next month. No further yeast issues persisted. As of August 2021, she was symptom-free for 4-months, and a CBC was back to normal (hemoglobin 13.4/hematocrit 40.0), with a normal CMP, thyroid functions, and hemoglobin A1c. Although frequent relapses occurred using dapsone combination therapy at doses of 50 mg to 100 mg per day during several courses of treatment, after 4 days of HDDCT, no further relapses occurred during the following year.

One year post-HDDCT, March 2022, she remained in full remission functioning at 100 to 110% normal. She felt better than she had been in the past several decades, as underlying fatigue and cognitive difficulties which had plagued her for her entire lifetime, were much better.

### 2.2. Case 2.2

This 29-year-old white male presented to our medical office in May 2018 with a past medical history significant for Asperger’s syndrome, ADHD, anxiety and depression, dyslipidemia, elevated homocysteine levels, chronic variable immune deficiency (CVID), testosterone deficiency, leaky gut with food allergies, and mold toxicity. He had a long history of chronic illness, starting with multiple tick bites when he was six years old living in Massachusetts. His health problems started shortly afterwards, when he developed significant fatigue, cognitive difficulties, along with anxiety and depression. He saw multiple medical providers over the next 23 years, but no cause for his problems could be found. His multiple specialists included general practitioners and several psychiatrists, who noted a poor clinical response to SSRIs and many other psychiatric medications. At the time of his initial visit, he was on a combination of venlafaxine, chlordiazepoxide HCl (Librium), aripiprazole (Abilify), and methylphenidate ER (Concerta) for his mood and concentration problems, along with exogenous testosterone replacement (clomiphene, testosterone gel). This combination improved his energy and stamina, decreased anxiety, and helped him to be more productive. After more than 20 years of illness, laboratory testing for tick-borne illness that was ordered by an integrative psychiatrist demonstrated exposure to Lyme disease and *Bartonella*, with active inflammation (increased CRP, homocysteine). He had not yet had treatment for his tick-borne illness when he first saw us during his initial visit. His symptoms at his initial presentation included moderate to severe fatigue, limited stamina, low libido, joint stiffness, neck stiffness, paresthesia’s in the extremities, poor concentration and executive functioning, anxiety and depression, and chronic insomnia, only getting between 4 and 6 h of sleep several nights per week.

A physical examination was significant for a well-developed well-nourished white male in no apparent distress. He was 6 feet tall, 255 pounds, afebrile, with a sitting blood pressure of 133/90 and pulse rate of 89 bpm (regular), which became 124/96 with a pulse rate of 105 bpm (regular) standing for 10 min. Family and social history, and a review of systems was essentially negative except for polydipsia and polyuria, male baldness pattern alopecia, diffuse acne, and some skin changes with horizontal striae across his shoulders, consistent with possible bartonellosis.

A review of lab testing revealed a CDC positive IgM Western Blot (23 kDa, 41 kDa positive), positive 39 kDa on a Lyme IgG Immunoblot (MDL Laboratories, Hamilton Township, NJ, USA), positive *Borrelia* EliSpot (4, >3 positive), positive *Bartonella* EliSpot (2, weakly positive), negative *Babesia* FISH, history of intermittently elevated liver functions, low vitamin D, elevated estradiol, and occasionally elevated blood sugars (non-fasting). He had not been officially diagnosed with nonalcoholic steatohepatitis (NASH) despite his elevated BMI and intermittently elevated liver functions, although this was confirmed later on during testing through our office. A low-level anti-mitochondrial antibody (AMA) was seen on follow-up test, and consultation with Hepatology ruled out primary biliary cirrhosis (PBC), but found that he had fatty liver, and was advised to go on a low carbohydrate diet and lose weight.

Other laboratory values that were reviewed showed that his vitamin and mineral levels were within normal limits, hemoglobin A1c was 5.2 (within normal limits), and he had normal thyroid functions, with intermittently low testosterone while on testosterone gel replacement (122, normal levels greater than 400). His immunoglobulin levels (IgA, IgM, IgG, IgG subclass 3) were below normal range, consistent with CVID, and a pneumococcal challenge in our office showed a poor antibody response. A 96 IgG food allergy panel through LabCorp revealed multiple food allergies that were consistent with leaky gut (barley, beef, casein, gluten, goat milk, oats, rye, wheat, whey). There was no evidence of heavy metal toxicity via venous blood analysis for mercury, lead, or arsenic, but aluminum levels in the plasma/serum were borderline elevated at 5 μg/L (environmental exposure 0–9). A urine mycotoxin test was sent to Real-time Labs to evaluate his mold exposure during his first visit and came back positive for 3/4 mold toxins (aflatoxins 1.556 ppb, normal range <0.8 ppb; trichothecenes 0.036 ppb, normal range <0.02 ppb; gliotoxins 3.357 ppb, normal range <0.5 ppb). He also sent off *Bartonella* testing through IgeneX Laboratory in California. A *Bartonella* Western Blot was negative, but *Bartonella* FISH testing was positive, confirming an active infection. Adrenal testing with a DHEA/cortisol salivary test through Labrix, revealed Phase 2 adrenal dysfunction, with cortisol in the low normal range throughout the day and night.

An antibiotic protocol was started on May 2018 for his Lyme and *Bartonella* exposure. He had previously taken tetracyclines in the past for acne without a clear clinical response and was initially started on cefuroxime axetil 500 mg PO BID and trimethoprim/sulfamethoxazole (Bactrim DS) one PO BID, while getting a baseline EKG to evaluate his QT interval and ability to take macrolides. His EKG returned within normal limits. Hydroxychloroquine 200 mg PO BID with Nystatin 500,000 units tablets two PO BID was also prescribed, with a sugar-free yeast-free diet, triple probiotics including *Saccharomyces boulardii* twice a day, with triple biofilm support (Biocidin, Stevia, oregano oil). Methyl protect (B6, B12, folic acid) was eventually given for his elevated homocysteine, and low dose naltrexone (LDN) was started at 2 mg HS, slowly working up to 4.5 mg HS, for his underlying inflammation.

The patient had an inadequate response to the above regimen and was tried on herbal protocols including monolaurin, berberine, A-Bart, and A-L (Byron White), and was rotated through several Lyme and *Bartonella* regimens over the next year, including several months of doxycycline 150 mg PO BID, rifampin 300 mg PO BID, pyrazinamide 2000 mg QD, and Zithromax 500 mg QD along with dapsone and methylene blue 50 mg PO BID. None of these regimens except double dose dapsone (DDDCT) had a significant effect on his fatigue, brain fog, or feeling of being constantly dehydrated (despite normal BUN/Creatinine values on a CMP), although the pyrazinamide also provided temporary benefit and symptom relief. During his first course of DDDCT, he had a significant improvement, and was the best he had felt in years and was able to return to school for the first time, but relapsed each time he was taken off of these medicines. He was, therefore, started on disulfiram, as a different persister medication, working up to 125 mg three times per week with doxycycline, rifampin, pyrazinamide, and methylene blue 50 mg PO BID, while lowering down doses of his psychiatric medication. As of November 2019, he continued to have resistant symptoms, and since he developed nausea on low doses of disulfiram, DSF and all antibiotics were stopped, getting a new baseline. Within two weeks of being off the antibiotics, he had a significant regression of joint pain, with increased fatigue and brain fog, so we restarted his antibiotic protocol. He also developed drenching night sweats during this time, and his testosterone was found to be extremely low at 122, so he was rotated off testosterone gels, and placed on clomiphene 50 mg, one half tablet three times per week with anastrozole 1 mg one time per week. His follow-up testosterone levels improved in the 400 s. His psychiatrist placed him on gabapentin 600 mg every eight hours for breakthrough symptoms, and he was referred to an immunologist to get IVIG for his CVID and resistant symptoms. His insurance company denied our request for immunoglobulin therapy, although his immunoglobulin levels gradually improved during the next year with his Lyme and *Bartonella* treatment. During this time, he was also taking phosphatidylcholine 3 g twice a day, glutathione (minimum 500 mg PO BID, higher doses for Herxheimer reactions) and a binder with charcoal, bentonite clay, and zeolite for his mold toxicity.

In February 2020, due to ongoing resistant symptoms, dapsone 100 mg per day was added back to his protocol of doxycycline, rifampin, and methylene blue 50 mg PO BID, along with hydroxychloroquine, leucovorin 25 mg PO BID, L-methyl folate 45 mg per day along with glutathione 1000 mg PO BID. Once this regimen was tolerated, double dose dapsone combination therapy (DDDCT) was reinstituted for a second time as it was one of the few effective regimens that he had taken, this time adding nitrofurantoin (Macrobid) 100 mg PO BID, for its published effects on *Bartonella* persisters. Unfortunately, as of April 2020, he felt a bit worse with increased fatigue and heaviness of his limbs on this protocol, although his sweats and neuropathy improved. The protocol was, therefore, DC’d. He continued on nystatin, probiotics, biofilm support, glutathione, N-acetyl-cysteine (NAC), and alpha lipoic acid, and during his follow-up consultation on July 2020, after being off antibiotics for 3-months, he again became incapacitated with severe fatigue, sweating, hair loss, moderate brain fog, with mild to moderate neuropathy. We sent off repeat levels of mycotoxins to evaluate their role in his resistant illness, and restarted him on Plaquenil, doxycycline 100 mg PO BID, rifampin 300 mg PO BID, pyrazinamide 2000 mg once a day, dapsone 50 mg/day, methylene blue 50 mg BID, and disulfiram 62.5 mg once a week, slowly working up the dose. His psychiatrist also began to further lower the dose of his venlafaxine during this time, due to lack of efficacy, and ongoing use of methylene blue, and he began to taper his Abilify, complaining of some clenching/discomfort and electrical sensations as doses were decreased, which eventually resolved. Ropinirole 1 mg HS was added to his protocol, which helped his insomnia and symptoms of RLS.

By November 2020, the patient was functioning at 5% of normal, due to severe fatigue, limited stamina, pain, and neurocognitive difficulties, so double dose dapsone combination therapy (DDDCT) was restarted for a third course, since it was the only combination therapy to date with a significant positive effect (post-Herxheimer reactions). It was taken with doxycycline, rifampin, pyrazinamide, methylene blue, and high dose glutathione (1000 mg twice a day), as he worked up the dose of double dose dapsone (200 mg/day). A new biofilm protocol was added for resistant symptoms, using cinnamon/clove/oregano oil capsules twice a day instead of oregano oil, along with compounded peppermint oil (200 mg) in capsules, with Stevia and Biocidin to address resistant biofilms. Zithromax, 250 mg PO BID, pulsed 4 days in a row per week was added once he tolerated the prior regimen. By the end of December 2020, the antibiotic protocol was stopped, and one month later, at the end of January 2021, he again noticed clinical improvement post-DDDCT, with increased energy, cognitive functioning, and less muscle stiffness. All laboratory testing remained WNL. Hemoglobin/Hematocrit was 12.6/38.4 on DDDCT and increased to 15.8/48.1 off medication with a normal CMP, and previously elevated inflammatory markers (CRP, ANA) decreased to normal. He evaluated his symptom improvement as 25% improved from his baseline, although he was still clinically disabled, functioning at only 30% of normal. A Lyme Immunoblot and *Bartonella* FISH were resent to IgeneX laboratory, with repeat hormone and mycotoxin testing. His testosterone levels improved on clomiphene and anastrozole to 639 ng/dL, thyroid function were WNL, but mycotoxin levels had increased despite treatment (Aflatoxins, 4.001 ppb, increased from 2.096; trichothecenes 0.206 ppb, increased from 0.009; gliotoxins 5.737 ppb, increased from 0.310; and zearalenone levels returned positive at 1.183). Lyme Immunoblots remained stable (there was slight increased banding at 23 kDa, Osp C, but no other new *Borrelia*-specific bands), *Babesia* FISH was negative, and a *Bartonella* FISH returned positive for a second time in 3 years on in March 2021, despite aggressive treatment with multiple intracellular medications and biofilm agents during the past several years. One month off antibiotics, he again began to relapse again with severe fatigue, brain fog, heavy limbs, low libido, hot flashes, and RLS, functioning at 5% of normal. He was, therefore, restarted on a slightly different Lyme and *Bartonella* protocol with minocycline 100 mg PO BID, clarithromycin 500 mg PO BID, pulse rifabutin 150 mg, two tablets twice a day, two days a week, with disulfiram, which was restarted at 62.5 mg once a week, working up to 62.5 mg/day over the next two months.

Since the patient had previously reported becoming ill in moldy environments, and had recently moved into a new apartment, an Environmental Relative Moldiness Index (ERMI) test through Mycometrics laboratory (Mycometrics, Monmouth Junction, NJ) was performed, testing up to 36 fungal species, in two groups. Group one was related to water damage and was positive for high levels of *Chaetomium globosum* (35) and *Eurotium (Asp.) amstelodami* (300) [less than 5 is considered normal] with an ERMI value of 3.19 [less than 2 is acceptable]. High *Chaetomium* levels can produce mycotoxins, coming from water-damaged carpets, windows, baseboards, or soil from indoor plants. The patient was, therefore, instructed to leave his apartment and have mold remediation done, and he was restarted on 3 g of phosphatidylcholine twice a day, glutathione 500 mg PO BID and a binder (G.I. Detox) with charcoal, clay, and zeolite, two hours away from all medication and supplements. During this time, he noted severe Herxheimer reactions restarting disulfiram at low dose (consistent with an active infection), and he was instructed to slowly increase the dose to 250 mg/day as tolerated. During the next 3-months, although there was less fatigue and improved mental clarity out of his mold-contaminated apartment, he otherwise denied any significant improvement in his clinical symptoms. As his neuropathy continued to flare up, even beginning with low dose disulfiram (62.5 mg/day), the DSF was stopped and he continued on minocycline 100 mg BID, clarithromycin 500 mg BID, and pulsed rifabutin 150 mg, two twice a day, two days per week, with prior doses of hydroxychloroquine, nystatin, biofilm agents (cinnamon/clove/oregano oil, peppermint oil), and probiotics remaining at the same dosage. Pyrazinamide (PZA) 2000 mg/day was added in August 2021, and after two weeks on PZA, he was instructed to go on high dose pulsed dapsone combination therapy (HDDCT), at 200 mg PO BID × 4 days for his resistant symptoms, along with increasing doses of methylene blue (working up from 50 mg BID to 100 mg BID) with Leucovorin 75 mg/day and L-methyl folate 45 mg/day. He was then instructed to stop all treatment and evaluate his clinical response one month later.

During the September 2021 consultation, one month post-HDDCT, the patient reported tolerating the regimen without difficulty. He reported Herxheimer reactions while on high dose dapsone, with a flare up of his fatigue and brain fog, which were improving off the protocol. He only reported using the PZA for 4 days during his course of HDDCT. Clinical symptom improvement included a significant decrease in neuropathy and improved sleep. Mitochondrial support with CoQ10 boosted energy, and with a strict low carbohydrate diet, he also lost 25 lbs., feeling much better at a lower weight. The patient was unable to get regular lab work done during his first course of HDDCT (he was out of the US traveling at the time). He repeated the HDDCT in early October 2021, for a second four-day course of medication. He had remained on hydroxychloroquine 200 mg PO BID, minocycline 100 mg PO BID, clarithromycin 500 mg PO BID, rifabutin 150 mg, two tabs PO BID, two times per week, and nystatin 500,000 units two tabs PO BID in between courses of dapsone, along with biofilm support with cinnamon/clove/oregano oil capsules twice a day, peppermint oil twice a day (for nausea and extra biofilm support), and grapefruit seed extract two capsules twice a day for the round body forms of *Borrelia burgdorferi*. He added in a four-day pulse of pyrazinamide 2000 mg PO QD and dapsone 200 mg PO BID with methylene blue 50 mg PO BID at the beginning of October, and by the end of the month, he had a significant improvement in his fatigue and brain fog. He stated that the second pulse of dapsone and PZA was easier to tolerate than the first one. His neuropathy steadily improved, and he was now back home in California in a new apartment that was mold-free. He had no difficulty tolerating the regimen of medications and support supplements.

Since the patient had an encouraging response to pulsed high dose dapsone therapy, which was well tolerated, without any signs of elevated methemoglobin levels (no significant blue hands, blue lips, shortness of breath) with a recent positive *Bartonella* FISH test, he was instructed to finish a third pulse of HDDCT with dapsone and PZA. He remained on bovine immunoglobulins and spore base probiotics for his leaky gut, along with his usual probiotics, avoiding sensitive foods, and received his second dose of COVID vaccination during this time, with only a mild flareup of symptoms. Amlodipine 5 mg once a day was added to his regimen for continued elevated blood pressure. By December 2021, the patient finished his third, four-day pulse of HDDCT, taken one month apart. His baseline functioning increased from 25% to 50 to 60% of normal (his usual functioning varied between 5 and 10% of normal), which is the highest it had been in the last several decades. He noted increased energy and stamina, significant improvement in cognition with less brain fog, improvement in his underlying neuropathy (which was previously severe, and was now mild), and mild, intermittent hot flashes. There were no Herxheimer reactions with the last pulse protocol, implying a significantly decreased load of persister/biofilm forms, confirmed by a repeat *Bartonella* FISH test which returned negative for the second time in three months. He was, therefore, instructed to stop all antibiotics once he finished the HDDCT protocol and do one month of mitochondrial regeneration with ATP 360 (Researched Nutritionals, glycosylated phospholipids) three once a day, Co-Q10 400 mg twice a day, acetyl-L-carnitine 1000 mg twice a day, and NADH 5 mg once a day. He noticed a definite improvement with mitochondrial support with Co-Q10 after one month, which improved his energy and ability to concentrate. His adrenal function with DHEA/cortisol levels were rechecked, which was improved from baseline studies, going from Phase 2 to Phase 1 adrenal dysfunction. A repeat Lyme Immunoblot that was performed in February 2022 through IgeneX laboratory revealed no new *Borrelia*-specific bands, negative *Babesia* PCR and FISH testing, and a second negative *Bartonella* FISH test. Two tests were positive for active Bartonella, 2018 and 2021, and were now negative X 2 since he completed three pulses of the HDDCT protocol. Repeat mycotoxin testing (Real-time Labs) also showed improvement, with lower levels of ochratoxins (1.295 ppb, not present, decreased from 1.512), aflatoxins (0.682, not present; decreased from 4.001), gliotoxins (2.166 ppb, decreased from 5.737), and zearalenone (0.57 ppb, equivocal, decreased from 1.183), with trichothecenes still remaining in the slightly elevated range (0.256 ppb, compared to 0.206). Follow-up blood tests showed a stable CBC post-dapsone (white blood cell 3.7, 0.1 below range; H/H 13.9/40.4), normal CMP, total testosterone of 639 ng/dL, and low normal estradiol levels (improved on anastrozole).

As of March 2022, the patient’s health had improved significantly compared to the past several decades. He was continuing to improve until he had mold exposure again while he was living in a hotel while his apartment was being renovated. This caused a regression of symptoms with increased fatigue, brain fog, and neuropathy. Once he left the hotel, he began feeling better again, as long as he was avoiding all moldy environments and strictly avoiding sensitive foods, sugar, and alcohol. He was instructed to continue on his mold detoxification protocol, using far infrared saunas when available to assist detoxification. This was the first time in years that the patient was able to remain off antibiotics for Lyme and *Bartonella*, after three consecutive pulses of HDDCT, while treatment of several overlapping MSIDS variables, especially mold, leaky gut and food sensitivities, hormone replacement (low T), and mitochondrial support all helped contribute to his clinical improvement.

### 2.3. Case 2.3

This 52-year-old white male with a past medical history significant for benign prostatic hypertrophy (BPH) and several surgeries (2 arthroscopic knee surgeries, left shoulder surgery, tonsillectomy, wisdom teeth removal) was in good health until 2004 when he developed a first-degree AV block, along with mild to moderate fatigue, numbness of his left hand, knee pain, back pain with muscular spasms, severe memory and concentration problems, headaches, vertigo, irritability, and insomnia. His initial Lyme titers were negative, and, after seeing several physicians, 6-months later he was eventually diagnosed with Lyme disease, babesiosis, and suspected bartonellosis (Fry Labs, Scottsdale, AZ, USA) positive hemo-bartonella). He initially had one year of IV antibiotics including doxycycline, ceftriaxone, Vancomycin, as well as Clindamycin and IV azithromycin, along with one year of intramuscular benzathine penicillin injections with rotations of oral atovaquone, atovaquone/proguanil, tinidazole, and levofloxacin. All antibiotics were helpful with his symptoms, but he continuously relapsed each time he was taken off antibiotics. A neuropsychiatric evaluation in 2005 revealed significant problems with processing speed which was 25% of normal for his age. He took 6-years of antibiotics until coming off all anti-infective agents in 2011. During this 6-year timeframe he developed multiple pulmonary emboli (2008) with an associated pneumonia, and required anticoagulation (enoxaparin, warfarin) along with pacemaker placement for long pauses (4–6 s) and tachyarrhythmias on his EKG. When the patient came to us in 2012, he had been off antibiotics for 6-months and all of his symptoms were significantly relapsing. He complained of severe cognitive difficulties with decreased memory and concentration, reversing letters and numbers, as well as joint stiffness in the morning, numbness of his left hand (consistent with possible median nerve involvement), mild headaches, occasional night sweats, moderate fatigue with good and bad days, rare dizziness, early awakening, and irritability. His wife described significant snoring, without an evaluation for obstructive sleep apnea (OSA).

Social history, family history, and a review of systems were otherwise unremarkable except for intermittent periods of hypertension, BPH, and easy bruising while on warfarin (Coumadin). His medications included methylphenidate (Concerta) 36 mg/day, lisdexamphetamine (Vyvanse) 50 mg per day, and escitalopram (Lexapro) 10 mg per day for his mood disorder and poor memory/concentration, along with warfarin 15 mg QD for the history of multiple pulmonary emboli, with a multivitamin and minerals. A physical examination revealed a well-nourished, white male in no apparent distress. He was 6′1″ tall, 247 pounds, afebrile, respirations 12 per minute with normal vital signs. A physical examination was unremarkable except for a mild increase in nasal turbinates, and ochre dermatitis of the lower extremities. All cranial nerves were grossly intact and sensory exam was normal to vibration and light touch. A CT of the head was reported as within normal limits.

Initial laboratory testing revealed an IgM Western Blot with a weak 23 (Osp C) and 41 kDa bands, and an IgG Western Blot with a weak 31 kDa (Osp A) and 93 kDa bands, with negative co-infection testing for *Babesia* (negative titers, negative *Babesia* FISH test), *Ehrlichia*, *Anaplasma*, *Bartonella* (negative titers for *Bartonella henselae* and *Bartonella quintana* with a negative VEGF), as well as negative testing for tularemia and rickettsial infections including *Coxiella burnetti*. Epstein-Barr virus titers showed evidence of an old infection, along with negative titers for CMV and West Nile virus. He was found to have a low testosterone varying between 109 and 215 ng/dL (normal range 348–1197), low DHT (15 ng/dl, normal range 30–85), intermittently low blood sugars (glucose 64, normal range 65–99) with a borderline elevated hemoglobin A1c at 5.8% consistent with hypoglycemia and metabolic syndrome (normal range 4.8–5.6), low 25 hydroxyvitamin D (13.1 ng/mL, normal range 30–100) and multiple IgG food allergies (milk, corn, eggs, wheat, beef). His immunoglobulin levels were within normal limits except for IGA deficiency (80 mg/dL, normal range 91–414) and IgG subclass 2 deficiency (235 mg/dL, normal range 242–700). Vitamin levels (B12, folic acid, methylmalonic acid) and thyroid functions were all within normal limits, as were all autoimmune markers (ANA, rheumatoid factor) except for a positive anti-myelin antibody. A DHEA/cortisol salivary adrenal test revealed Phase 2 adrenal dysfunction with low cortisol levels at noon, in the evening and nighttime (0.4–1.25 nmol/L, normal range 2.1–15.7), a low DHEA at 107 pg/mL (normal range 137–336) and a 6-h urine DMSA challenge was positive for elevated levels of mercury (26 micrograms/g creatinine, normal range less than 3) and elevated levels of lead (14 micrograms/g creatinine, normal range less than 2). Inflammatory markers were elevated with a high C-reactive protein (CRP) at 3.2 mg/L (normal range less than 1–3), high transforming growth factor beta (TGF-beta) at 21,674 pg/mL (normal range 3465–13,889), and elevated complement studies with a C4a at 2935 ng/mL (normal range less than 2830). He also had an abnormal lipid panel with hypertriglyceridemia at 407 mg/dL (normal range less than 150), a total LDL direct of 124, and elevated Lp(a) at 15 mg/dL (normal range less than 10). He was, therefore, placed on K2 D3 5000 IUs per day, a low carbohydrate diet with fenofibrate 145 mg once a day, amlodipine 5 mg QD with valsartan 160 mg QD for his hypertension and hyperlipidemia, along with cefdinir 300 mg PO BID, probenecid 500 mg BID, azithromycin 250 mg PO BID, and Nystatin 500,000 units 2 PO BID for his chronic Lyme disease. Eventually atorvastatin 10 mg QD was given for his elevated cholesterol and multiple cardiac risk factors. Ashwagandha, rhodiola, B vitamins, and low dose hydrocortisone 5 mg in the morning, 2.5 mg at 2 pm were given for adrenal support, along with probiotics.

The patient had a positive response to the oral antibiotics, changes in diet, and adrenal support, with a decrease in his systemic symptoms, including improvement in his cognition. Over the next several years, within several weeks of stopping his oral antibiotics his memory and concentration would worsen significantly as would his fatigue and pain. As of May 2015, he was on amoxicillin 875 mg, 3 tablets PO BID with probenecid 500 mg PO BID with a normal peak and trough level, but still suffered with debilitating symptoms. He was, therefore, rotated to minocycline 100 mg PO BID and rifampin 300 mg PO BID which improved his energy and physical symptoms with mild improvement in his cognition. His depression increased during this time due to marital stress, and bupropion (Wellbutrin) 150 mg PO BID was added to his escitalopram along with trazodone 200 mg HS for depression/insomnia along with memantine (Namenda ER) 14 mg per day for ongoing cognitive impairment. None of these changes however had a significant effect and the patient was sent for psychological therapy, with 15 mg of l-methyl folate added to try and augment the effectiveness of his SSRI. Due to ongoing neck, arm, and right-hand pain, a CT of the cervical spine without contrast was performed, which showed degenerative changes, a narrow spinal canal, broad-based posterior disc bulging at C2/C3 with mild central stenosis and moderate foraminal narrowing, with moderate to severe bilateral foraminal stenosis (right greater than left) at C5/C6 and C6/C7. An EMG was performed of the right upper extremity which ruled out an associated radiculopathy or carpal tunnel syndrome.

Due to ongoing resistant symptoms, including fatigue, knee pain, and severe cognitive difficulties, the patient was started on dapsone January 2016, adding it to minocycline and rifampin, working up to 50 mg/day after his G-6-P-D levels returned WNL. He noticed an immediate improvement in knee pain which resolved after three weeks on dapsone, with less fatigue and slightly improved cognition, although word finding problems were still significant. Laboratory testing was stable with an H&H of 13.3/40.7 (from 15.3/45.9) with 25 mg of l-methyl folate. He described this regimen as being almost as good as his intramuscular/IV therapies. He continued on this therapy for several months, and as of March 2016, as resistant symptoms persisted, the dapsone dose was increased to 100 mg/day. This caused a significant Herxheimer reaction with an increase in brain fog and more emotional swings, however his home situation had worsened with increased stress and further lack of sleep, complicating his clinical picture. He was taken off all antibiotics which helped to decrease his Herxheimer reaction, and several months later was rotated to pulse azithromycin 250 mg PO BID and cefuroxime axetil 500 mg BID, four days in a row per week, along with sulfamethoxazole/trimethoprim (Septra DS) one PO BID. Unfortunately, several months later, cognitive difficulties significantly worsened with decreased memory and processing speed, along with increased dizziness, neck pain, and increased neuropathic symptoms in his lower extremities. A neurological consultation was obtained and no new diagnoses were provided apart from the patient’s chronic tick-borne illness, with a myelogram reported as normal. Doxycycline sustained-release 150 mg PO BID was, therefore, added to minocycline sustained-release 90 mg QD for better CNS penetration, along with Zithromax 250 mg PO BID four days per week and Bactrim DS one PO BID. Hydrocortisone dosing was slightly increased for significant fatigue, which helped improve his energy. As of January 2017, although there was a minor improvement in cognitive function, without any further dizziness, rifampin was re-added to his protocol due to resistant symptoms. In June 2017, he was functioning at 85% of normal, and continued to improve overall, feeling the best he had felt in quite a few years. His cognition and energy levels had been better, and he was much more physically active at his job. Biofilm support was added with Stevia Extract 15 drops twice a day, but several months later, joint pain in his knees relapsed again with resistant insomnia, with a significant worsening of cognition. All antibiotics were, therefore, temporarily held to get a new baseline, and the patient was restarted on doxycycline 150 mg PO BID, clarithromycin 500 mg PO BID, and Bactrim DS one PO BID. He was then seen in our medical office in February 2018. He had traveled to Maine in the interim and had two engorged tick bites with a physician diagnosed *erythema migrans* rash despite being on doxycycline. He complained of increased fatigue, knee pain, chest pain, and shortness of breath with a dry cough post-tick bite. He went to the emergency room and was ruled out for a heart attack or pneumonia with a negative CT scan of the chest, normal EKG, and follow-up negative exercise stress test. The patient was, therefore, placed back on doxycycline 200 mg PO BID, rifabutin 150 mg PO BID, and gradually increasing doses of dapsone, doing his first course of double dose dapsone combination therapy (DDDCT), working up to 100 mg PO BID for one month’s time in the summer of 2018. His CBC remained stable at 12.5/38.1 with normal blood methemoglobin levels (0.1%, 2.4%, 3.1% on successive blood tests, normal levels less than 5%).

As of July 2018, although he felt better post-DDDCT with improved energy, cognition, and pain after having a significant Herxheimer reaction (old Lyme symptoms returned with temporary vertigo, shakes, and knee swelling), off all antibiotics for 2-months, within several weeks, underlying symptoms began to relapse again. We, therefore, retested *Bartonella* titers, VEGF, and a Quest laboratories tick-borne panel, and sent him to an ENT physician for purulent discharge seen on physical examination in the right ear canal. All tick-borne testing including PCR’s for *Ehrlichia*, *Anaplasma*, *Babesia microti*, *Borrelia burgdorferi*, and *Borrelia miyamotoi* were negative, and he was placed back on antibiotics for an otitis media. In November 2018, after a trial of minocycline, Zithromax, rifampin, and pyrazinamide for a suspected resistant *Bartonella* infection post-DDDCT, he reported an improvement in symptoms from 55 to 70% of normal, but fatigue, pain, and cognition increased in severity with Herxheimer reactions, and he was again temporarily taken off antibiotics. Once the patient was stable, doxycycline, hydroxychloroquine, rifabutin, and dapsone were restarted, and as of December 2018, the dapsone protocol helped his symptoms with improvement in fatigue, memory and concentration, and less shoulder pain, but increased knee pain. A repeat IgM and IgG Lyme immunoblot January 2019 revealed decreased *Borrelia*-specific bands over time (23 kDa [OspC], 39 kDa) but a *Bartonella* IgM Western Blot was performed for resistant symptoms (despite negative local testing) and returned indeterminate positive with exposure to *Bartonella vinsonii* sub-species, with a *Bartonella* FISH turning positive, confirming an active *Bartonella* infection. This was despite multiple IM, IV, and oral intracellular antibiotic combinations during the past 15 years. Pyrazinamide 2000 mg per day was, therefore, re-added to his regimen for active *Bartonella*, and the dapsone dosage was increased from 50 to 100 mg per day. As of April 2019, the patient again described a worsening of symptoms with fatigue and short-term memory problems, although his clinical picture was exacerbated by working long hours (12 to 14 h per day) with a maximum of 6 h of sleep per night, interrupted by frequent awakening. A total of 2-months later, on minocycline 100 mg PO BID, hydroxychloroquine 200 mg PO BID, rifabutin 150 mg PO BID, pyrazinamide 2000 mg per day, and dapsone 100 mg per day, the patient was feeling much better, improving from 60 to 85% of normal. His energy had improved, cognition was significantly better, and insomnia and joint pain had improved. High dose glutathione (2000 mg per day) also helped improve brain fog, and heavy metal testing showed improvement over time on a 6-h urine DMSA challenge, with mercury levels decreasing from 26 to 14, and lead levels decreasing from 14 to 10. Methemoglobin levels were stable on methylene blue 50 mg twice a day, with no significant anemia on dapsone combination therapy.

In June 2019, we discussed adding disulfiram to the dapsone antibiotic protocol as a second persister drug for his active Lyme and *Bartonella*. He stopped Stevia and Biocidin (biofilm agents with alcohol) and added serrapeptase 2 twice a day and monolaurin one scoop a day to his oregano oil for biofilm support. He was instructed to gradually work up the dose of disulfiram (DSF) to 250 mg twice a day over the next month, avoiding all alcohol, and holding the dose for any significant increases in Herxheimer reactions, or neuropathy. By August 2019, he was on full dose disulfiram for 2 ½-months. He initially flared, with an increase in his underlying symptoms after three weeks on DSF, and high dose glutathione (2000 mg) and detoxification support (N-acetyl-cysteine) helped to decrease Herxheimer reactions. While on minocycline 100 mg PO BID, rifabutin 150 mg PO BID, pyrazinamide 2000 mg per day, dapsone 100 mg per day, methylene blue 50 mg PO BID, hydroxychloroquine 200 mg PO BID, and disulfiram 250 mg PO BID, he was functioning at approximately 80% of normal and had plateaued. He then was instructed to again stop all antibiotics to get a new baseline, remain on high dose folic acid and probiotics with biofilm agents, while doing 1–2-months of mitochondrial support with ATP fuel (glycosylated phospholipids), CoQ-10,400 mg twice a day, 1000 mg of acetyl-L-carnitine twice a day, and NADH 5 mg once a day. By December 2019, 3-months off disulfiram and dapsone, again, his symptoms were relapsing with significant cognitive difficulties, word finding problems, and forgetting conversations. His worsening cognition rapidly happened within three weeks off the protocol. The patient tried to remain off antibiotics until March 2020, except for cefuroxime axetil 500 mg PO BID when he was going for a total knee replacement. He felt that disulfiram was previously helpful when it was added to his other antibiotics, helping with fatigue and pain, although he described increased depression while on it. The patient was, therefore, kept off disulfiram, and restarted on minocycline 150 mg twice a day, dapsone 50 mg PO BID, rifampin 300 mg PO BID, azithromycin 250 mg PO BID, methylene blue 50 mg PO BID, leucovorin 25 mg BID, and 15 mg of l-methyl folate, for *Borrelia burgdorferi* and *Bartonella* persister/biofilm forms. As of December 2020, although there was some benefit with this protocol, knee pain was relapsing with exercise/walking and memory/recall issues continued to be problematic. Disulfiram was then re-added to his protocol, working up to a lower dose of 125 mg per day, trying to avoid the psychological side effects, and low dose naltrexone (LDN) 4.5 mg PO QAM was also added for immune/inflammatory support. Repeat tick-borne testing was sent to IgeneX Laboratory which showed a slight increase in his 23 kDa (OspC) on a Lyme IgG immunoblot, and a *Babesia* test that returned positive (*Babesia* FISH test) confirming active infection. Atovaquone/proguanil (Malarone) 250/100 mg, 2 PO BID were added with a high-fat meal and the disulfiram dosage was increased to 187.5 mg for resistant babesiosis.

The patient remained on this regimen until June 2021 when disulfiram was discontinued. He remained stable at 85% of normal and completed another sleep study for ongoing insomnia, since he was only sleeping 5 to 6 h per night, contributing to his underlying fatigue and cognition. Mild-moderate sleep apnea was found and he was placed on CPAP, which improved his insomnia. By August 2021, remaining on minocycline, pulse azithromycin and rifampin, with dapsone 50 mg per day, he felt the dapsone combination therapy continued to help his symptoms, and glutathione continued to help reduce his brain fog. The *Babesia* symptoms were gone after several months of Malarone, dapsone, disulfiram, and methylene blue. He had plateaued at 80% of normal with moderate fatigue, moderate memory, and concentration problems, along with moderate joint pain in his knees, neck, and shoulders. Due to ongoing resistant symptoms, in November 2021 the patient worked back up to 250 mg of disulfiram and started on his first course of high dose dapsone combination therapy (HDDCT) with a seven-day course of disulfiram 250 mg per day, nystatin 500,000 units two PO BID, minocycline 150 mg PO BID, hydroxychloroquine 200 mg PO BID, rifampin 300 mg, 2 PO BID, dapsone 100 mg BID × 3 days followed by 100 mg, 2 PO BID for 4 days with methylene blue 100 mg PO BID. This was taken with three biofilm agents (cinnamon/clove/oregano oil PO BID; serrapeptase 2 capsules twice a day; monolaurin one scoop per day) and three different probiotics for G.I. support.

By January 2022, roughly one month after completing his first round of HDDCT, the patient reported feeling the best he felt in a long time. The knee pain, brain fog, and word finding problems had significantly improved (at least 10% better) but there was no change in his fatigue. CBC was stable at 11.6/36.5 with low methemoglobin levels of 0.6% and 1% during therapy. We, therefore, repeated a second round of HDDCT with 250 mg of disulfiram. He tolerated the regimen well without any associated nausea or vomiting, or gastrointestinal problems (no yeast, no diarrhea). Methemoglobin levels continued to remain in normal range (2.2%, 1.7%) without any signs of symptoms of methemoglobinemia (blue hands/blue lips, shortness of breath). There were no Herxheimer reactions after the second course of HDDCT, implying a significant lowering of biofilm/persister forms, and his energy, knee pain, and cognitive functioning significantly improved. The patient, therefore, stopped all of his antibiotics early January 2022 and remained on Nystatin for the next month with probiotics, tapering down methylene blue. He repeated one month of a mitochondrial regeneration protocol post-HDDCT, and 2 *Bartonella* FISH tests returned negative by 2020 after multiple rounds of intracellular antibiotics with dapsone combination therapy. During his last visit on April 2022, the patient had remained in remission for 3-months after two rounds of HDDCT with 250 mg of disulfiram. He had increased energy (he felt normal for the first time in years), no relapse of knee pain, and improved memory/concentration. He still had some mild word finding issues which continued to be helped with high dose glutathione, implying the need for ongoing detoxification support, but they were significantly better from his initial visit 10 years before. This was the first time that there had been no relapse of any of his underlying Lyme and tick-borne symptoms during the past 20 years.

## 3. Materials and Methods

We closely examined 25 charts of adult patients who completed the HDDCT protocol. A total of 22 of the 25 patients (88%) had previously done one to two courses of DDDCT and relapsed post-treatment. We assessed co-infection status, age, length of illness, and response to treatment, i.e., self -reported improvement in Lyme symptoms, and whether there was remission, or lack of response to the HDDCT protocol. Remission was defined as the resolution of all active tick-borne symptoms for 3-months or longer. All 25 patients in our retrospective chart review met the criteria for a clinical diagnosis of Lyme disease supported by a physician-documented *erythema migrans* (EM) rash and/or positive laboratory testing, including a positive ELISA/enzyme immunoassay (and/or C6 ELISA), immunofluorescent antibody (IFA), Centers for Disease Control and Prevention (CDC) positive IgM and/or IgG Western Blot (WB), PCR, *Borrelia*-specific bands (23, 31, 34, 39, 83/93) on a WB [14], and/or positive ELISpot (lymphocyte transformation test (LTT)). The majority of patients that were enrolled had previously undergone one or more treatments with disulfiram and/or DDDCT and had either failed or had an inadequate response to prior antibiotic therapy, and/or had relapsed with persistent symptoms after stopping anti-infective therapy.

All the patients on this protocol had signed informed consent forms that outlined the proposed benefits and the potential risks of our study; patients volunteered to enroll in this high dose pulsed dapsone study at our medical center based on our prior research illustrating the benefit of dapsone combination therapy in the treatment of CLD/PTLDS [18,30,39], and on the drug’s documented action on “persister” bacteria in biofilms [34].

None of the patients that were enrolled had a sulfa allergy or G-6-P-D deficiency, in order to minimize the possibility of allergic reactions or severe hemolytic anemia secondary to dapsone. Prior to beginning HDDCT, the patients were required to have a hemoglobin greater than 12 mg/dL, no active bleeding disorders, and no contraindication or allergies to any of the medications or supplements. The side effects of dapsone were explained, including potential rashes, Herxheimer reactions, anemia, and methemoglobinemia [40]. The patients were asked to obtain a baseline methemoglobin level before starting pulsed HDDCT to ensure that there were no significant baseline elevations due to genetic variations or other medication interactions [41,42]. A complete blood count (CBC), comprehensive metabolic profile (CMP) with electrolytes, kidney, and liver function as well as methemoglobin levels were obtained before, during and 3–4 weeks after the completion of the protocol.

The potential side effects of HDDCT were addressed using medication and nutritional supplements/anti-oxidants with anti-inflammatory effects, including glutathione, N-acetyl cysteine, and alpha lipoic acid, which block NFKappa-B [20], as well as high dose folic acid (Leucovorin, L-methyl folate) and oral methylene blue. Glutathione and methylene blue both decrease methemoglobin levels [41,43] and high dose folic acid helps to minimize anemia from inhibition of bacterial synthesis of dihydrofolic acid by dapsone. Folinic acid supplementation has been shown in prior studies to help limit myelosuppression, gastrointestinal toxicity, nephrotoxicity, and neurotoxicity that can result from high dosages of folic acid antagonists [44]. High dose probiotics including *Saccharomyces boulardii* were used to help prevent antibiotic associated diarrhea [45,46].

In order to minimize potential interactions with methylene blue, including serotonin toxicity [47], patients on psychiatric medications, especially a selective serotonin reuptake inhibitor (SSRI) or monoamine oxidase inhibitor (MAOI), were asked to taper their psychiatric medication, and/or consult with their psychiatrist before increasing the dose of methylene blue to a final dose of 100 mg twice a day. Dietary restrictions also were instituted, avoiding foods high in tyramine, which potentially can produce a hypertensive crisis in the presence of MAOIs and/or higher dose methylene blue. Patients were given a list of foods to avoid during the protocol, which included: aged cheese, aged chicken or beef liver, air-dried sausage and similar meats, avocados, beer, and wine (in particular, red wine), canned figs, caviar, fava beans, meat tenderizer, overripe fruit, pickled or cured meat or fish, raisins, sauerkraut, shrimp paste, sour cream, soy sauce, and yeast extracts [48].

Several patients took disulfiram (DSF) in combination with HDDCT, if they had previously failed to have an adequate clinical improvement with either drug regimen used alone or in combination. The patients signed a consent informing them of the potential benefits and risks of DSF [32,33], including increased fatigue, brain fog/cognitive dysfunction, worsening psychiatric symptoms, liver function abnormalities, and/or increased neuropathy [49]. They were instructed to stop DSF immediately if there was any worsening in underlying neuropathic symptoms, and to contact our office immediately if any severe adverse effects were noted. The dose of DSF that was used in prior clinical studies ranged from a lower dose DSF (250 mg or less) to a higher dose DSF (500 mg/day) [32,33]. The majority of patients who took DSF in our study used doses of 250 mg/day or less, to minimize the potential adverse effects and Herxheimer reactions. Disulfiram doses were slowly increased over time, increasing the dose by 62.5 mg every 1–2 weeks until reaching the target dose, which was based on efficacy and tolerance. The patients were instructed to use sodium bicarbonate or freshly squeezed lemons and/or limes to alkalize the body if Herxheimer reactions resulted from killing off of *Borrelia* [50,51], along with N-acetyl cysteine, alpha lipoic acid, and high dose glutathione. [52,53,54]. These nutraceuticals help to decrease inflammation through their effect on blocking NFKappa-B, lowering inflammatory cytokine production [55,56]. Apart from the above nutritional supplements and dietary recommendations, the DSF group were also instructed to avoid ingestion and exposure to alcohol, to minimize gastrointestinal complications including nausea and/or vomiting [57], a known side effect of DSF toxicity.

Although both dapsone and disulfiram have both demonstrated some efficacy against resistant biofilm/persister forms of *Borrelia burgdoferi* [30,58,59], three different biofilm agents were used during the HDDCT protocol to improve efficacy. These included Stevia [37], Biocidin [38], and essential oils including oregano, cinnamon, and clove [35,36]. If patients were on DSF, monolaurin and serrapeptase were substituted as biofilm agents [60,61] instead of using Stevia and Biocidin, since they do not contain any alcohol. Grapefruit seed extract was also used along with hydroxychloroquine for their effects against the round body forms of *Borrelia* [62,63].

An Institutional Review Board approval was not required for this research since this was a retrospective review of a convenience sample of patient charts. A convenience sample of 25 patient charts were chosen for inclusion in our study out of a patient population of 45 patients that were offered HDDCT (some patients had not yet started HDD CT or were still on the protocol at the time of publication). Table 1 below lists the medications and nutritional supplements that were used before, during and after HDD CT.

## 4. Results

Of 25 participants, 10 were male and 15 were female. The age range was between 18 and 68 years old (M = 46.72, SD = 13.522). A total of 52% were less than 50 years old, and 48% were greater than 50 years old. A total of 40% of patients (N = 10) had been ill for greater than 20 years, 40% (N = 10) had been ill between 10 and 20 years, 20% (N = 5) had been ill between 5 and 9 years, and no patients had been ill for less than 5 years in duration. A total of 28% of patients (N = 7) had a history of EM rashes, and therefore met the criteria for PTLDS. Three out of 25 patients (12%) had at least one co-infection, 14 (56%) had two co-infections, seven (28%) had three or more co-infections, and one patient had no co-infections. A total of 9 patients (36%) were *Babesia microti* antibody positive, 5 patients (20%) were *Babesia duncani* antibody positive, 11 patients (44%) were *Babesia* FISH positive, 5 patients (20%) were *Ehrlichia* antibody positive, 2 patients (8%) were *Anaplasma* antibody positive, 16 patients (64%) were *Bartonella* antibody positive (*B. henselae*, *B. quintana*), 1 patient was *Bartonella* PCR positive, 4 patients (16%) were *Bartonella* FISH positive, and 3 patients (12%) had evidence of an elevated VEGF, an indirect marker of active *Bartonella*. The co-infection status and treatment results are presented in Table 2.

Treatment results: A total of 21 of 25 patients (84%) showed improvement of their tick-borne symptoms, and one patient had a temporary worsening of symptoms post-therapy, secondary to a severe Herxheimer reaction. A total of 32% (8/25) had a resolution of all active Lyme symptoms post-treatment for 3-months or longer even if there was evidence of prior active co-infections, including *Babesia* and *Bartonella*. A total of 3 out of 7 patients (43%) with an EM rash and history of PTLDS remained in remission, and the remainder, 57%, showed a mild-moderate improvement in underlying symptoms above their baseline functioning. Among 11 patients who were *Babesia* FISH positive, three (27%) remained in remission, six improved (55%), and two (18%) had no change in sweats, chills, or flushing or dyspnea. Among 19 patients (76%) with a history of *Bartonella* exposure, 15/19 improved (79%), and four patients (21%) had no change in symptoms. Among 8 patients with proof of active *Bartonella* infection (PCR positive, FISH positive, and/or elevated VEGF), 3/8 (38%) remained in remission and 5/8 (62%) improved their underlying symptomatology. Further examination of those patients revealed that among those who were *Bartonella* FISH positive, two out of four went into remission (50%) and the other two improved their underlying symptomatology by 10% to 20% above baseline functioning. The largest symptom improvement that was seen among this cohort that were treated with HDDCT were cognition 17/25 (68%), joint and muscle pain 16/25 (64%), fatigue 15/25 (60%), headache 9/25 (36%), neuropathy 5/25 (20%), insomnia 5/25 (20%), sweats/chills/flushing (20%), palpitations 2/25 (8%), tinnitus 2/25 (8%), dyspnea 1/25 (4%), and visual “snow” 1/25 (4%).

The number of courses of DDDCT and HDDCT, taken with or without disulfiram (DSF) or prior to using HDD CT was also evaluated to determine the effect on symptom resolution and remission rates. A total of 16 patients (64%) had taken doses of DSF ranging from 62.5 mg per day to 250 mg BID (time frames ranging from 3–4-months to 18-months). Among those who had done DSF prior to HDD CT, 50% (8/16) reported some improvement in symptoms after a Herxheimer reaction, including improved fatigue, pain, headaches, cognition, and sweats, without any significant long-term side effects. Disulfiram alone was insufficient to keep them in remission however, as three patients were *Bartonella* FISH positive post-DSF, and two patients were *Babesia* FISH positive post-DSF. Two patients out of eight (25%) that went into remission did HDDCT with DSF (200–250 mg) simultaneously. Among those patients who took DSF prior to HDDCT, 62.5% (5/8) went into remission, although prior use of DSF alone with or without lower dose dapsone combination therapy (50–200 mg) was insufficient to keep them symptom-free until HDDCT was added to their protocol. One course of HDDCT was sufficient in 75% (6/8) of patients who went into remission, two courses were needed in one patient (case 2.2, positive *Bartonella* FISH), and three courses were needed in one patient (case 2.3, positive *Babesia* FISH and *Bartonella* FISH).

Age and length of illness was also evaluated to determine the effect on the treatment outcome. Among 12 out of 25 patients less than 50 years old, 4 patients went into remission, 7 patients improved, and 1 patient had no change in symptoms. In those 13 patients that were greater than 50 years old, 4 patients were in remission and 6 improved, while three had no change in clinical status. Regarding the length of illness, 10/25 patients were ill for 20+ years and three went into remission, four improved, and three had no change in symptoms; ten patients were sick between 10 and 20 years, and 3 remained in remission, six improved and one had no change in symptoms; five patients were ill between 5 and 9 years, and two went into remission, while three improved; and finally, 0 patients were sick for 1–4 years. The four patients who did not improve on the HDDCT protocol had all been exposed to *Bartonella* (*B. henselae*, and/or *B. quintana*), 2/4 (50%) were *Babesia* FISH positive with evidence of exposure to *B. microti* (1:320+), and had multiple overlapping MSIDS variables interfering with their clinical improvement, including significant mold toxicity (2/4 patients), mast cell activation (1/4), evidence of exposure to heavy metals (2/4), detoxification problems with low glutathione (2/4), *Candida* (2/4), food allergies/sensitivities (2/4), adrenal dysfunction (2/4), hypoglycemia (2/4), hypothyroidism (3/4), chronic variable immune deficiency (CVID) (1/4), evidence of an overactive immune system (1/4), and resistant insomnia (1/4). The MSIDS variables that were found among the 25 chronic Lyme/PTLDS patients which were interfering with treatment outcome is presented in Figure 1.

Laboratory values were monitored before and after HDDCT. White cell counts and platelet counts remained WNL, as did kidney and liver functions (transaminases). One patient (1/25) had a temporary elevation in bilirubin to 6.0% on dapsone which resolved within one month off therapy [40]. Hemoglobin and hematocrit values decreased temporarily on HDDCT, with a mean drop in hemoglobin of 1.75 g during the four-day pulse (N = 10; range 0.6–3.1), returning to normal levels within one month off the protocol while remaining on high dose Leucovorin and L-methyl folate. Methemoglobin levels ranged between 0.3% and 23.9% (N = 17) while on methylene blue and glutathione, with a mean increase in methemoglobin of 7.97% during the four-day period. The highest methemoglobin level was in a patient who was not taking oral glutathione but was on glutathione suppositories instead (500 mg QID). All methemoglobin levels returned to normal range (<2%) when checked three to four weeks off treatment. The majority of patients tolerated HDDCT well, with no clinical signs of methemoglobinemia, although three patients with methemoglobin levels that were greater than 10% had transient blue hands and blue lips with shortness of breath, which quickly resolved off dapsone [64]. No evidence of serotonin syndrome was seen among 40% (10/25) of patients who were taking methylene blue and were previously on psychiatric medicine (including bupropion, paroxetine, escitalopram, duloxetine, doxepin, trazodone, Seroquel, buspirone, cyclobenzaprine, imipramine, and/or mirtazapine). All patients were ruled out for Glucose 6 Phosphate Dehydrogenase (G-6-P-D) deficiency prior to using dapsone and methylene blue (MB), instructed of the potential side effects of MB (hypersensitivity reactions, anaphylaxis, hemolytic anemia, serotonin syndrome, hypertensive crisis, and syncope) as well as the possibility of extremity pain, urine discoloration, fecal discoloration, hot flashes, dizziness, hyperhidrosis, skin discoloration, nausea, headache, syncope, chest pain, diaphoresis, vomiting, back pain, abdominal/bladder pain, and photosensitivity. Those patients who were on an SSRI or other psychiatric medication with a potential interaction were instructed to lower their doses of psychiatric medicines and/or speak to their psychiatrist prior to using HDDCT. No patient had any hypertensive crisis, and the most significant side effect that was noticed during HDDCT was nausea (N = 7) and vomiting (N = 4), which was improved using ondansetron 8 mg Q 8 h before or during treatment. Finally, no patients had antibiotic-associated diarrhea using three different high dose probiotics including *Saccharomyces boulardii*.

## 5. Discussion

A chart review of 25 patients who were at least 3-months out after completing HDDCT revealed the importance of several factors. Age did not appear to be a major variable in determining the treatment outcome in this small retrospective study, as equal numbers of individuals above and below 50 years old achieved remission. Regarding the length of illness, most patients had been ill for a long period of time, and a length of illness that was greater than 20 years in duration appeared to lead to worse remission outcomes. Among four patients who had no response to HDDCT, three had been ill for 20 or more years. This is consistent with data that were collected from other published studies that tracked patient progress over time [65], highlighting that patients who experience delayed diagnosis may be more likely to develop chronic Lyme disease [66], where severe complications can occur if treatment is delayed [67].

Lyme and associated tick-borne disorders are increasing worldwide, with prior studies indicating that at least 10 to 20% of individuals go on to suffer from CLD/PTLDS [68]. In 2021, the largest prospective study to date of adults with physician-confirmed Lyme Borreliosis (LB) was performed in Europe, and the prevalence of persistent symptoms in LB patients was found to be higher at 27.2% [69]. Studies that were done in the US indicate that the number of patients suffering from PTLDS may be even greater. At six months post-EM rash, a John Hopkins University study found that 36% of patients reported new-onset fatigue, 20% reported widespread pain, and 45% complained of ongoing neurocognitive difficulties [70]. These were the top three symptoms that were reported by these patients and also the three that most improved after the HDD CT protocol. Based on research that was done in 2019, where the cumulative prevalence of PLTDS in the United States is estimated to be high and continuing to increase, affecting close to two million individuals [4], it is essential to explore innovative solutions for this expanding public health crisis.

There is considerable medical debate regarding the etiology, incidence, and severity of CLD/PTLDS [71], with treatments remaining controversial. PTLDS is characterized by incapacitating fatigue, pain, and neurocognitive dysfunction that persists for more than 6-months, with intermittent or constant symptoms that are often subjective and varied in nature [72]. The diagnosis is often made based on the exclusion of other conditions [5]. Horowitz et al. proposed a clinical definition for CLD/PTLDS, using a 16-point precision medical model that is known as MSIDS [20]. In that model, multiple overlapping causes of inflammation and downstream effects driving symptomatology were noted in chronic LD patients with or without a history of an EM rash. A patient symptom survey and a retrospective chart review of 200 patients with chronic Lyme disease/PTLDS [20] identified those variables on the MSIDS model with the greatest effect on regaining health. The results indicated that dapsone combination therapy at doses of 100 mg per day combined with a tetracycline and rifampin decreased the severity of eight major Lyme symptoms [18], and found that multiple sources of inflammation (other infections such as *Babesia* and *Bartonella*, reactivation of viruses, immune dysfunction, autoimmunity, food allergies/sensitivities, leaky gut, mineral deficiencies, environmental toxins with detoxification problems, and sleep disorders) along with the downstream effects of inflammation (POTS/dysautonomia, autoimmunity, mitochondrial dysfunction, hormonal dysfunction, liver dysfunction, and neuropsychiatric symptoms) can all affect chronic symptomatology [20]. These were the same variables that affected the treatment outcomes in these 25 patients on HDDCT.

As per Figure 1, among our cohort of chronically ill patients, the top MSIDS variables interfering with long-term health outcomes were hormonal dysfunction (low adrenal function), with several individuals having HPA axis dysregulation with low testosterone and/or hypothyroidism, depression/anxiety/trauma, resistant insomnia, persistent *Babesia*, persistent *Bartonella*, EBV reactivation, food sensitivities with leaky gut, with or without *Candida* and mast cell activation, environmental toxicity (mold, heavy metals) with detoxification problems, immune dysfunction (CVID, subclass deficiency), autoimmunity and/or mitochondrial dysfunction. Based on three prior dapsone studies with a total of 340 chronically ill Lyme patients, which also evaluated MSIDS variables [18,20,30,39], we believe that it is important to address all underlying MSIDS factors contributing to chronic illness. A total of 6 out of the 16 MSIDS factors have also now been found in long COVID, i.e., post-acute sequelae SARS-CoV-2 infection (PASC), another chronic fatiguing, and musculoskeletal illness with neuropsychiatric manifestations. These include viral persistence [73], persistent inflammation [74], autoimmunity [74], EBV reactivation [75], mitochondrial dysfunction [76], and POTS/dysautonomia [77]. Considering the millions of individuals that now suffer with long COVID, with the prevalence of residual symptoms being about 35% in patients that were treated for COVID-19 on an outpatient basis, and around 87% among cohorts of hospitalized patients [78], an investigation of the 16-point MSIDS model could be considered in this patient population to screen for other overlapping sources of inflammation and downstream effects which have not yet been explored. This requires shifting the paradigm from a ‘one cause/one disease’ model to a multifactorial healthcare model. The MSIDS model has been shown to help define, diagnose, and treat not only CLD/PTLDS but also other chronic illnesses [20,21,22]. These could include long COVID, Myalgic encephalomyelitis (ME), and fibromyalgia. At a minimum, at least one in seven COVID-19 patients is symptomatic at 12 weeks with PASC, with complaints of fatigue, post-exertional malaise, and cognitive dysfunction seen by month six [79], which are part of the same symptom complex affecting millions of individuals worldwide who suffer from chronic medical conditions including M.E., fibromyalgia, PASC, and chronic Lyme disease/PTLDS. These conditions all share similar clinical characteristics of chronic fatigue, pain, cognitive dysfunction, insomnia with unrefreshing sleep, mood disorders, and orthostatic intolerance/POTS [80,81,82]. In Lyme disease, the migratory aspect of the pain is, however, a hallmark finding [83] which is only found in six other medical diseases [83], which helps to differentiate these four clinical syndromes. Exploring the biochemical mechanisms of other diseases with similar clinical characteristics may help to elucidate the etiology and effective treatment for long COVID. We are presently awaiting the start of a randomized, controlled clinical trial on the 16-point MSIDS variables in long haulers at the University of California, Irvine [84].

Prior to using pulsed HDDCT (200 mg of dapsone PO BID × 4 days) and applying the MSIDS diagnostic and treatment model for the treatment of CLD/PTLDS, we had published on the efficacy of double dose dapsone combination therapy (DDDCT) for the treatment of CLD/PTLDS [39]. In that study, lower doses of dapsone combination therapy, i.e., 100 mg PO BID for one month, preceded by four weeks of gradually increasing doses of dapsone combination therapy (25 mg × 1week; 50 mg × 1 week; 75 mg × 1 week; 100 mg × 1 week in combination with a tetracycline and rifampin) was effective in helping chronically ill Lyme and tick-borne patients go into long-term remission for one year or longer. This included patients who had been sick for more than two decades. A retrospective chart review of 40 patients that were undergoing DDDCT therapy demonstrated tick-borne symptom improvements in 98% of patients, with 45% remaining in remission for one year or longer if there was no evidence of active co-infections [39]. A total of 7 out of 12 patients (58%) with an EM rash and history of PTLDS also remained in remission, and the remainder, 42%, showed a moderate improvement in underlying symptoms (37%) above their baseline functioning [39]. In this study, only two patients had not previously done DDDCT, i.e., case 2.1, and another patient who could not previously tolerate high dose persister drugs secondary to severe Jarisch–Herxheimer reactions. The majority of participants, i.e., 92% (23/25) had already done DDDCT with ongoing resistant symptoms, which is why they were offered high dose pulse dapsone combination therapy.

Prior culture studies using dapsone alone and in combination with other intracellular antibiotics (tetracyclines, rifampin, azithromycin) had shown that dapsone as a single drug and in combination with doxycycline +/− rifampin as well as doxycycline + rifampin + azithromycin had the most significant effect in reducing the mass, viability, and protective mucopolysaccharide (MPS) layers of *B. burgdorferi* biofilm [34]. *Borrelia* persister bacteria in biofilms have been determined to be a significant cause of inflammation [27] that are known to be a major factor in driving chronic symptomatology in Lyme disease patients with CLD/PTLDS [82], and higher doses of dapsone in culture (50 μM vs. 10 μM concentrations) were found to be more effective than lower doses in reducing the mass, viability, and MPS of *Borrelia* biofilm [34]. This might explain, at least in part, dapsone’s clinical efficacy that was seen in recent DDSCT trials and in our present trial of HDDCT. A similar finding with another ‘persister’ drug for Lyme disease, disulfiram, was found to be more effective at keeping patients in remission if it was taken at higher doses [33]. In that study, 36.4% of patients who completed one or two courses of “high-dose” therapy enjoyed an “enduring remission”, defined as remaining clinically well for ≥6-months without further anti-infective treatment, and high-dose patients were significantly more likely to achieve enduring remission [33]. We found similar results in our study. DDDCT at 100 mg PO BID was more effective than single dose dapsone (100 mg), and HDDCT (200 mg PO BID × 4 days) with one to three courses being more effective than DDDCT (200 mg PO BID × 28 days) in helping to keep certain patients in long-term remission as long as the majority had already done one or more courses of DDDCT and all abnormal MSIDS variables were addressed. One patient who failed DDDCT and was on DSF intermittently for up to 18-months, at doses ranging from 250 mg per day to 250 mg PO BID, improved her underlying symptoms of fatigue, headaches, and mood disorder. She felt DSF was more effective for her symptoms of CLD/PTLDS (EM+) but continued to relapse off DSF. She took one course of HDDCT after coming off DSF and went into remission for 3-months or longer. Combining two persister drugs such as dapsone (DDS) or DSF, or rotating them, was effective in this patient who did not have an adequate response to one prior course of DDSCT. Of the eight patients who went into remission, 5/8 (62.5%) took DSF in varying dosages either before or during HDDCT, and 4/8 (50%) had prior exposure to *Bartonella*. Future studies should evaluate the efficacy of combining persister drugs to improve long-term remission for CLD/PTLDS and associated infections.

The primary differences between DSF and HDDCT for CLD/PTLDS are the length of time on the drug regimen, the potential side effects, and efficacy. In prior DSF clinical trials, patients took one to two courses of DSF ranging from 6 weeks to greater than 16-months [32,33]. In DDDCT and HDDCT, drug regimens averaged 8–10 weeks [39]. There were also statistical differences between the high-dose and low-dose DSF patients in the incidence of adverse reactions as well as achieving enduring remission. The patients who took DSF in the high dose group had common adverse reactions including fatigue (66.7%), psychiatric symptoms (48.5%), peripheral neuropathy (27.3%), and mild to moderate elevation of liver enzymes (15.2%) with 36.4% achieving long-term remission [33]. Patients who took DDDCT frequently had Herxheimer reactions with a temporary increase in fatigue, pain, and ‘brain fog’, but had a long-term benefit and reduction in eight major Lyme symptoms, including fatigue, pain, peripheral neuropathy, and psychiatric symptoms [39]. A total of 39 of 40 patients (98%) on DDDCT showed improvement of their tick-borne symptoms. No patients had a worsening of symptoms post-therapy and 45% remained in long-term remission or one year or longer, with 58% of patient with a history of PTLDS remaining in remission [39]. DDDCT followed by HDDCT is, therefore, a shorter, more effective treatment than DSF in the CLD/PTLDS population, with potentially fewer side effects, but some patients may benefit from using both persister drugs in their treatment course to lower the load of persister bacteria increasing inflammation.

The four major side effects of HDDCT that are addressed in the dapsone protocol, using high dose folic acid (L-methyl-folate, Leucovorin), methylene blue, glutathione, and antioxidants can be expressed as ‘Do no ‘H.A.R.M.’, i.e., Herxheimer reactions, anemia, rashes, and methemoglobinemia. Herxheimer reactions are due to inflammatory cytokine release with spirochetal killing [50]; anemia, secondary to inhibition of folic acid metabolism, or hemolysis due to G-6-P-D deficiency [40]; rashes, if significant sulfa sensitivity exists; and methemoglobinemia, due to increased levels of oxidative stress with decreased oxygen-carrying capacity [85]. Patients had been ruled out for significant sulfa sensitivity and G-6-P-D deficiency prior to using dapsone, so no rashes were noted. Although none of our 25 patients had a history of sulfa sensitivity, our prior experience using dapsone in patients who have a history of mild rashes to sulfamethoxazole/trimethoprim (not severe reactions, i.e., no history of anaphylaxis or Stevens–Johnson syndrome) is that the majority are able to tolerate dapsone, another sulfa drug, without any allergic reactions. If any question exists regarding allergic history, and the risk/benefit is determined to be in favor of using dapsone or disulfiram (both sulfa drugs), dermatological/allergy consultation should be obtained to confirm sulfa sensitivity via skin testing with consideration of using H1/H2 blockade (i.e., cetirizine/famotidine). Dapsone’s (DDS) half-life is much shorter than DSF (28 h vs. 7–14 days) [86], ensuring that if significant side effects occur (Herxheimer reactions, rash, anemia, methemoglobinemia), they also resolve more quickly.

The risk of significant anemia during HDDCT was minimized using high dose folinic acid (75 mg PO BID) with high dose L-methylfolate (45 mg PO BID) and mean drops in hemoglobin of 1.75 g during the four-day pulse (N = 10; range 0.6–3.1) were noted, with hemoglobin levels returning to normal levels within one month off the protocol while remaining on tapering doses of high dose folic acid (Leucovorin and L-methyl folate). The administration of glutathione precursors (NAC 600 mg PO BID) and Alpha lipoic acid 600 mg PO BID with glutathione 2000 mg PO BID served to not only block NFKappaB and help decrease Herxheimer reactions and inflammatory cytokine production [56,87,88], but also support detoxification and anti-oxidant activity along with methylene blue. Methylene blue (MB) has several functions, primarily to help lower methemoglobin levels while on dapsone [85], apart from its effects on the biofilm/persister forms of *Borrelia* and *Bartonella* [89,90]. Methemoglobin levels ranged between 0.3% and 23.9% (N = 17) while on methylene blue and glutathione, with a mean increase in methemoglobin of 7.97% during the four-day period, and all methemoglobin levels returned to normal range (<2%) when they were checked three to four weeks off treatment. Higher dosing of folic acid, methylene blue, and glutathione were used in those patients on HDDCT compared to DDDCT [39], to help prevent an increase in Herxheimer reactions, anemia, and methemoglobinemia, with favorable outcomes. The majority of patients tolerated HDDCT well, with no clinical signs of methemoglobinemia, although three patients with methemoglobin levels that were greater than 10% had transient blue hands and blue lips with shortness of breath, which quickly resolved off dapsone. The highest methemoglobin level that was seen (23.9%) was in a patient who was not taking oral glutathione 2000 mg PO BID but was on glutathione suppositories instead (500 mg QID), which could account for the discrepancy. The symptoms of methemoglobinemia are proportional to the fraction of methemoglobin in the blood, and a normal methemoglobin fraction is roughly 1% (range, 0–3%), with symptoms occurring at higher levels of methemoglobin. Most patients with a methemoglobin level of <10% have no symptoms, although patients with underlying diseases such as Lyme disease and associated co-infections may notice more symptoms at lower levels, i.e., headaches, mild shortness of breath. Levels of 10–20% are usually associated with slight discoloration of the skin, e.g., pale, gray, bluish tint of the skin, and levels of 20–30% are oftentimes associated with anxiety, headache, tachycardia, and lightheadedness [91]. Our patient with a methemoglobin level of 23.9% was relatively asymptomatic, although other patients with levels that were greater than 10% have noted headache and shortness of breath with changes in skin color. Significant reactions that were secondary to elevated levels of methemoglobinemia did not occur during the use of HDDCT, as they have been noted to occur at methemoglobin levels that are greater than 30% with life threatening reactions reported at levels of 50% or greater (arrhythmias; altered mental status, delirium, seizures, coma; profound acidosis) [91]. None of our patients had methemoglobin levels that were greater than 30% or noted any of these severe side effects, as the mean increase in methemoglobin was 7.97%. For patients that are sensitive to elevated levels of methemoglobin that are greater than 10% or are felt to be at risk for higher levels due to underlying medical conditions (asthma, emphysema, heart disease, etc.), apart from temporary use of supplemental oxygen (none of our patients required home oxygen), extra antioxidant support can be used. This would include using higher doses of NAC (1200 mg PO BID), glutathione (increasing dosage to 2000 mg TID) and NADH [43], ascorbic acid, and Vitamin E [92] as well as cimetidine [93], which helps to decrease methemoglobin levels and improve the therapeutic/toxic ratio of dapsone in patients on chronic dapsone therapy [94]. Methylene blue (MB) was also used to lower methemoglobin [94], gradually increasing dosages four days before using HDDCT to ensure tolerance. The most common side effect that was noted with MB was the sensation of bladder irritability, which resolved promptly off treatment. Methylene blue was used at a dosage of 100 mg PO BID during HDDCT, and patients were instructed to taper off any medications which could interact with MB beforehand, including selective serotonin uptake inhibitors (SSRI’s) to avoid any potential side effects such as serotonin syndrome. Methylene blue may contribute to serotonin syndrome if it is used in combination with other serotonergic drugs such as SSRIs, selective norepinephrine reuptake inhibitors (SNRIs), MAOIs (monoamine oxidase inhibitors), and tricyclic antidepressants (TCAs) due to its MAOI activity [47]. A comprehensive drug interaction check should, therefore, be performed prior to using MB to ensure safety [95], and Lexi-Interact and Epocrates, have been shown to provide some of the most accurate software programs, based on the programs’ sensitivity, specificity, and positive and negative predictive values in detecting drug-to-drug interactions (DDIs)s [96]. Patients were also instructed to avoid tyramine-containing foods while on MB to avoid potential hypertensive effects [97].

In this study, the risk/benefit ratio of using four days of higher dose dapsone combination therapy was evidenced by 21 of 25 patients (84%) showing improvement of their tick-borne symptoms, and 32% (8/25) having a resolution of all active Lyme symptoms post-treatment for 3-months or longer even if there was evidence of prior active co-infections, including *Babesia* and *Bartonella*. A total of 3 out of 7 patients (43%) with an EM rash and history of PTLDS remained in remission, and the remainder, 57%, showed a mild-moderate improvement in their underlying symptoms above their baseline functioning. Short-term high dose pulsed dapsone had improved efficacy in those that were failing lower doses of dapsone and was effective in some patients with resistant *Babesia* and *Bartonella*. Prior studies using DDDCT [39] showed that among seven patients who were *Bartonella* FISH positive, although none of them achieved long-term remission, 2/4 (50%) of our patients in this HDD CT study who were *Bartonella* FISH positive remained in remission after 2-3 courses of HDDCT (cases 2.2, 2.3). Similarly, in our prior study of DDDCT, among six patients who were *Babesia* FISH positive, three (50%) remained in remission, and the other three patients improved (50%). In this study, 3/11 patients (27.3%) who had previously failed DDDCT and multiple *Babesia* therapies, including clindamycin [98], atovaquone and azithromycin [99,100], atovaquone/proguanil [101], tafenoquine [102], lumefantrine/artemether [103], cryptolepis, and artemisinin [104] went into remission with a resolution of malarial-type symptoms (sweats, chills, flushing, unexplained cough, and “air hunger”) after using 1–3 four-day courses of HDDCT.

*Bartonella* and *Babesia* can potentially be cured if they are treated early on in immune competent patients, although both organisms can lead to chronic persistent infections in CLD/PTLDS [18,105,106], ultimately leading to long-term resistant symptomatology in this patient population [18,39]. Persistence of *Babesia* and *Bartonella* after standard therapeutics was highlighted in the 2018 HHS Report from the Other Tick-Borne Diseases and Co-Infections Subcommittee that was presented to the Tick-Borne Disease Working Group [106]. Regarding babesiosis, resistance to atovaquone and azithromycin (A + A) as well as clindamycin and quinine (C + Q) have been attributed to the rapid emergence of drug resistance [107], and treatment failures with both of these standard babesia drug regimens have been reported in not only Lyme disease patients, but also immune competent and immunosuppressed patients [108,109]. Both dapsone and DSF have some efficacy in treating *Babesia* [18,30,32,33,39], but generally are not adequate to clear the infection. In our prior and present dapsone combination studies, these two co-infections were responsible for a large percentage of resistant symptoms [39]. This is consistent with other studies of chronically ill Lyme patients, showing the co-infections are the rule, not the exception in this patient population [109]. A big data sample of almost 4000 people that were diagnosed with Lyme disease from the MyLymeData patient registry by Johnson et al. reported coinfections with *Babesia* in 23% of individuals (21% without supporting lab tests) and *Bartonella* (19% with supporting lab tests, 23% without) [66]. We only counted positive co-infection data for those with laboratory confirmed *Babesia* and *Bartonella*, where 20/25 patients (80%) in our HDDCT study were positive by antibody testing for exposure to either *B. microti* or *B. duncani*, with 44% (11/25) being active for babesiosis by *Babesia* FISH. Sixteen out of 25 patients (64%) were also positive for exposure to *B. henselae* and/or *B. quintana,* with 16% (4/25) active by *Bartonella* FISH, 1/25 (4%) positive by *Bartonella* PCR, and 12% (3/25) active with an elevated VEGF, an indirect marker of active *Bartonella*. These co-infections were responsible for a large percentage of resistant symptoms. Only 33% (3/9) with exposure to *B. microti* went into remission, 50% (2/4) with exposure to *B. duncani* went into remission, and 27.3% (3/11) who were *Babesia* FISH positive went into remission. A similar finding was seen with *Bartonella*. Roughly 19% (3/16) with exposure to *Bartonella* by antibody testing went into remission, 33% (1/3) of VEGF positive patients achieved remission, 50% (2/4) *Bartonella* FISH positive patients went into remission, and one patient who was PCR positive for *Bartonella* failed to achieve remission.

Expanded direct testing by PCR and FISH, not relying solely on antibody testing, is important to determine the co-infection exposure rates in this tick-borne patient population, especially since patients with chronic LD/PTLDS patients have been shown to be immunosuppressed. In a prior study of 200 Lyme disease patients with associated co-infections who were treated with dapsone combination therapy, 7% had chronic variable immune deficiency (CVID), 20.6% had total IgG deficiency, and 85.5% had combined IgG subclass deficiencies [18,20]. Immune deficiency can lead to false negative antibody testing, or if a patient is on IV immunoglobulins (IVIG) or subcutaneous immunoglobulins (SQIG), false positive antibody testing can result due to pooled blood specimens from the general population. The *Babesia* FISH test detects *Babesia duncani* and *Babesia microti*, two common species that cause human infections in the USA [110], and the *Bartonella* FISH test by IgeneX detects bacteria of the Genus *Bartonella* including *B. vinsonii, B. berkhoffii, B. henselae*, and *B. quintana* in whole blood smears. Fluorescence in situ hybridization (FISH) is a complementary molecular tool for the clinical diagnosis of infectious diseases by intracellular and fastidious bacteria that can be difficult to diagnose through standard antibody testing [111], and our chronically ill patient population demonstrated high co-infection rates for *Babesia* and *Bartonella* by antibody, with proof of active co-infections by PCR, VEGF, and FISH testing. In eight patients, FISH testing was positive for either *Babesia* or *Bartonella* when antibody testing was negative, and in one patient PCR was positive when antibody testing was negative. In other Lyme patients with resistant symptoms including dermopathy (Morgellons’s disease), resistant *B. henselae* co-infection was found in 20% using PCR (blood and/or skin samples) and FISH (30% of skin samples positive) [112]. If we combine the data from our present study of 25 patients on HDDCT with the 40 chronic Lyme/PTLDS patients in our DDDCT who were also co-infected [39], only 35.5% (6/17) went into remission if there was proof of active *Babesia* by *Babesia* FISH, and only 18.2% (2/11) who were *Bartonella* FISH positive went into remission. There is, therefore, an urgent need to find effective treatments for both co-infections to help improve the clinical outcomes in this very sick cohort of chronically infected patients.

In two out of four *Bartonella* patients who went into remission for 3-months or longer, and were active for *Bartonella* via *Bartonella* FISH, we added additional treatments with two other intracellular antibiotics (azithromycin and pyrazinamide) along with an additional essential oil (peppermint) to attempt to achieve greater efficacy. Dapsone combination therapy uses a tetracycline (minocycline or doxycycline), rifampin (or rifabutin) with dapsone, along with hydroxychloroquine, nystatin, methylene blue, and a combination of three biofilm agents. Some of the biofilm agents that are routinely used in dapsone combination therapy contain essential oils [EO’s] (i.e., Biocidin and cinnamon/clove/oregano oil), which have been shown to have efficacy against the biofilm/persister forms of Bb [35,38]. Stevia does not contain EO’s but has been also shown to be effective against *Borrelia* biofilms [37]. Serrapeptase and monolaurin were substituted as biofilm agents in our treatment group if the patient was on DSF, to avoid exposure to alcohol, which would potentially increase acetaldehyde and gastrointestinal symptoms of nausea and/or vomiting [61,113]. We added peppermint oil in case 2.2 as he had two *Bartonella* FISH tests that were positive after years of prior treatment with multiple intracellular antibiotics, and peppermint oil has been shown to act as a biofilm agent, mediating quorum sensing and potentially lowering the minimal inhibitory concentrations (MIC) of certain antibiotics [114]. Prior studies had shown that *Bartonella* can form highly resistant dormant persister bacteria in biofilms [115], similar to Lyme disease [116,117], and that these essential oils can have high activity against stationary phase *Bartonella henselae* [115] as well against stationary phase *Borrelia burgdorferi*. We, therefore, regularly used a combination of several essential oils in our treatment protocol to address biofilm formation of Lyme and associated co-infections including *Bartonella* in order to try and improve efficacy. Whether the addition of peppermint oil to other EO’s can improve efficacy in treating resistant biofilm infections needs to be explored.

Other culture studies have identified drug combinations that were effective against *Borrelia burgdorferi* persisters in vitro including daptomycin, cefoperazone and doxycycline [23], sulfa drugs [118] including dapsone [25], as well as disulfiram, which was discovered using high throughput screening of FDA-approved drug candidates [58]. We, therefore, used combination drug regimens with dapsone (DDS) as our primary persister drug with or without DSF to address *Borrelia* persisters. Prior culture studies showed that dapsone alone was effective against the resistant biofilm forms of *Borrelia burgdorferi* [34], and that the triple combination treatment of dapsone + doxycycline + rifampin (52% residual viability) and quadruple combination of dapsone + doxycycline + rifampin + azithromycin (58% residual viability) treatments both at the higher 50 μM concentration were the most effective combinations (*p*-values < 0.01) against the resistant forms of *Borrelia burgdorferi* [34]. This four-drug antibiotic combination (doxy, rifampin, azithromycin, dapsone) using higher doses of dapsone in HDDCT was found to be effective for patient 2.2 who had Lyme and active *Bartonella*, when used with methylene blue and pyrazinamide (PZA). Two prior courses of DDDCT with lower doses of dapsone (100 mg BID) were insufficient to keep him in remission, which is why PZA, and azithromycin was added to his protocol. Pyrazinamide was previously published to have some efficacy against *Bartonella* [119] and combinations of azithromycin, rifampin, and methylene blue have also been published to have efficacy against persister forms of *Borrelia* and *Bartonella*. In prior culture studies looking at effective treatments against the persister forms of *Bartonella*, methylene blue, was also found to be an active drug against persister forms of *B. burgdorferi* [90] and was found to have good activity against stationary phase *B. henselae* [120]. Combinations of methylene blue/ rifampin as well azithromycin/methylene blue were the most active agents against the biofilm forms of *B. henselae* after six days of drug exposure [90]. These medications, therefore, have overlapping and potentially synergistic effects against multiple tick-borne infections including *Borrelia* and *Bartonella*. Larger controlled randomized trials of these antibiotic protocols are required to confirm efficacy for chronic persistent *Borrelia* and *Bartonella* infections.

Regarding *Babesia*, there have been very few new medications and published studies used for treatment-resistant infections, and persistent parasitemia after acute babesiosis has been reported since 1998 [105,121]. Concurrent babesiosis with Lyme disease is known to increase the severity and duration of illness [17], and only one medication, tafenoquine, has been recently published in a case study, as an effective treatment for relapsing babesiosis with clinical and molecular evidence of resistance to azithromycin and atovaquone (A + A) [102]. The standard seven-day treatment regimen of A + A is no longer adequate to treat a *Babesia* infection, due to mutations in the cytochrome B of *Babesia* parasites [107], and higher doses, longer treatment duration, and occasionally IV administration are required for severe infection [122]. Clindamycin and Quinine (C + Q) also may no longer be effective for severe babesiosis, requiring whole-blood exchange transfusion [98]. Both A + A and C + Q have significant adverse side effects, including, respectively, diarrhea and rashes, and tinnitus, diarrhea, and decreased hearing [100]. Novel therapeutics are, therefore, being explored, and tafenoquine, clofazimine, and endochin-like quinolones appear to be the most promising drug candidates [103], but no randomized, controlled trials have been performed to date, leaving chronically ill patients without good options. Considering the importance of persistent *Babesia* infection increasing chronic symptomatology among our cohort of chronically ill Lyme and co-infected patients, trials of novel *Babesia* medications and/or herbal therapies need to be prioritized, along with randomized trials of persister drug/biofilm regimens for chronic Lyme and *Bartonella* infections.

The emergence and spread of antibiotic resistance among pathogenic bacteria is a major healthcare issue in the treatment of infectious diseases, and about 80% of the infections that are caused by microorganisms have been determined to be biofilm based [123]. The survival of bacteria that are present in the biofilm takes place in the form of microcolonies, which are encapsulated in the extracellular polymeric substance (EPS) of the matrix, protecting them from antibiotics [124]. Increased antibiotic resistance has proven to be a common trait that is associated with biofilm bacteria [125], and until several years ago, we were unaware that *Borrelia* and *Bartonella* had biofilm/persister forms that could interfere with treatment, leading to long-term persistence. In the past several decades, university-based researchers have published on the ability of *B. burgdorferi* to change morphological forms in different environments [126,127], forming round bodies (S-forms, L-forms, cell-wall-deficient forms) [128,129], and metamorphosing into drug-tolerant persister and biofilm forms [24,26,117,130,131]. Persisters can occur in a broad range of bacterial infections, not just Lyme disease and Bartonellosis, where the infections can go dormant and relapse, leading to persistent infection [132,133,134,135]. As persister cells do not grow in the presence of antibiotics, they become a significant fraction of cells, without any genetic modification that make up these stationary phase cells in biofilms [136]. Stationary persister cells in biofilms potentially play a major role in antibiotic resistance and relapse of *B. burgdorferi*, based on culture studies confirming the efficacy of dapsone against the resistant biofilm form of Bb [34], and our positive clinical results improving eight major Lyme symptoms in three prior dapsone studies involving 340 patients [18,30,39]. Microbial biofilm formation has also potentially been shown to play a role in neuroborreliosis, with European strains of *Borrelia*, *B. afzelii,* and *B. garinii* [136,137]. Based on this emerging research, and the number of patients that are suffering from CLD/PTLDS and other associated tick-borne infections, along with the clinical success we are having using novel persister drug regimens in patients who have failed traditional antibiotic protocols, it is essential to confirm this research in larger, well-controlled, randomized trials. Prior NIH double-blind placebo-controlled studies which looked at the efficacy of longer treatment courses of antibiotics for CLD/PTLDS did not account for the presence of biofilm/persister forms of *Borrelia* [138,139,140], and also did not account for the role of chronic co-infections including *Babesia* and *Bartonella*, nor the role of MSIDS variables causing chronic illness. We are finding these variables to be essential factors in helping chronically ill Lyme and tick-borne patients get better and stay in remission. The other advantage to using one, or several pulses of our HDDCT preceded by DDDCT, is that these are oral, generic medication regimens that do not require IV therapy, and are only used for short periods of time (9–10 weeks). As antimicrobial resistance is becoming a major healthcare problem worldwide, using multiple intracellular antibiotics simultaneously as we did in HDDCT, with different mechanisms of action, can help to limit resistance. This is routinely done in persister drug regimens that are used for slow growing persister bacteria including *Mycobacterium tuberculosis* [132]. A variety of infections that are linked to biofilm production have now emerged worldwide, posing a global health threat, by rapidly increasing the statistics of antimicrobial resistance. Exploring the tenets of translational medicine, the efficacy of different essential oils and medications to treat resistant biofilm infections in Lyme and associated co-infections, may help us to design more effective treatments for other chronic biofilm diseases.

Our study has several limitations, as outlined below. Our cohort was too small to determine if adding disulfiram to dapsone prior or during HDDCT and using DSF in different dosages and lengths of treatment, significantly affected the treatment outcomes. Larger numbers of patients in well-controlled, randomized studies will be needed to evaluate the efficacy of HDDCT after using DDDCT with or without DSF in those suffering from chronic Lyme/PTLDS, with or without Bartonellosis and babesiosis. Our follow-up and operational definition of remission was also evaluated at a three-month interval, as some patients were just finishing a third course of HDDCT, so efficacy and relapse rates long-term is unknown. The efficacy of HDDCT was similarly evaluated in a very ill patient population with multiple overlapping medical problems, as noted in Figure 1, where up to 18 different variables on the MSIDS map were found to be present. It is possible efficacy will vary and potentially improve, in a cohort of patients who are not ill for 20+ years with so many overlapping sources of inflammation. As an example, mold toxicity, with or without multiple chemical sensitivity, was found in 9/25 patients (36%), and mycotoxin exposure has been linked to chronic fatigue syndrome (CFS/ME) [141], where some individuals with mold exposure had gliotoxins known to be immunosuppressive [142], potentially interfering with the treatment response. POTS/dysautonomia was found in 7/25 patients (28%). The symptoms of POTS include fatigue, dizziness, palpitations, anxiety, and cognitive difficulties [22,143] and may have influenced the treatment results. One patient with mercury toxicity (blood level greater than 25 μg/L, upper range less than 10) noted significant help with resistant tinnitus, fatigue, and cognition using HDDCT, but was simultaneously being chelated for his elevated level of mercury with low dose dimercaptosuccinic acid (DMSA) [144], so the treatment effects may have coincided. Multiple overlapping variables on the MSIDS map in this patient population thereby makes it difficult to get a full picture of the efficacy of HDDCT, and these variables will need to be accounted for in future studies. We also did not regularly evaluate patients for the presence of other tick-borne infections including hard tick relapsing fever, *Borrelia miyamotoi*, nor exposure to the deer tick (Powassan) virus, which are emerging tick-borne infections [145,146], which potentially can influence morbidity. Similarly, although there are multiple *Bartonella* species that are now found in ticks, it is unclear whether all of these are tick-transmitted pathogens, and we were unable to screen for the broad range of *Bartonella* species emerging in the scientific literature as potential pathogens [147,148]. Despite these challenges however, HDDCT appears to be a promising novel therapy, which should be explored with scientific rigor and randomized, controlled trials, as few effective options are now available for those that are suffering from CLD/PTLDS.

## 6. Conclusions

We are presently experiencing a global rise in tick-borne infections worldwide due to climate change [149,150] and the healthcare burden of expanding tick-borne diseases is significant in terms of healthcare cost, potential disability, and long-term suffering [8,151]. Only two persister drug regimens presently exist for the treatment of CLD/PTLDS, dapsone combination therapy and disulfiram, with two potential new treatments on the horizon, Azlocillin and Hygromycin A [152,153], both which require clinical evaluation. Since there is no established safe and effective cure for babesiosis and Bartonellosis, both important co-infections in our patients and in the chronic LD population [66], and no recent randomized clinical trials for Lyme disease in over a decade, it is essential that governmental agencies, university-based researchers, and clinicians in the field come together to help find cures for these diseases, as the burden of individuals that are exposed to Lyme disease and associated tick-borne infections continues to increase every year [154] with recent studies reporting an estimated global *Borrelia burgdorferi* sensu lato seroprevalence being 14.5% [155].

Our study points to the possibility that it is not necessarily the length of time on persister drugs such as dapsone, but rather the dose that may play a critical role in addressing resistant biofilm forms of *Borrelia burgdorferi*. An evaluation of this hypothesis and the efficacy of novel persister drugs such as dapsone and disulfiram in Lyme disease, as well as the role of the MSIDS model in chronic fatiguing, musculoskeletal illnesses with neuropsychiatric symptoms that are affecting large numbers of individuals across the globe, i.e., CFS/ME, FM, CLD/PTLDS, and long COVID, demands that we expand our research in these areas for the benefit of all who are suffering and have suffered with these chronic diseases, allowing us to provide hope, and potentially novel scientific solutions.

## Figures and Tables

**Figure 1 antibiotics-11-00912-f001:**
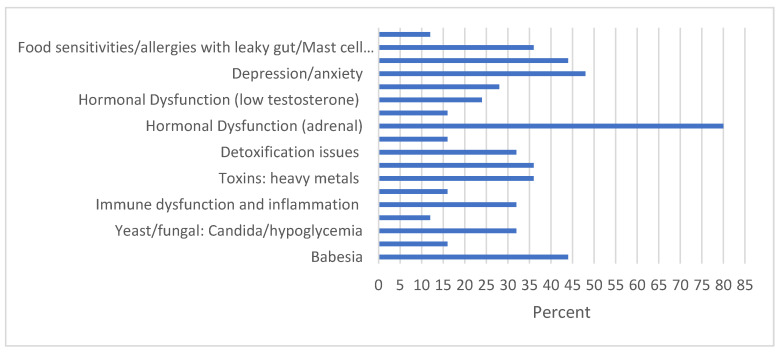
MSIDS variables in 25 patients with CLD/PTLDS: Abbreviations: Non-alcoholic steatohepatitis (NASH), mast cell activation disorder (Mast Cell).

**Table 1 antibiotics-11-00912-t001:** Medication and nutritional supplementation that were used during high dose pulsed dapsone combination therapy.

Time Frame on HDD CT	Medication	Nutritional Supplementation
Days 1 + 2 (pre-HDD CT)	Methylene blue 50 mg BID Continue prior medication for Lyme and associated tick-borne disease, other medical conditions	Triple biofilm agents (Biocidin, 2 sprays BID; Stevia, 15 drops BID; cinnamon/clove/oregano oil capsules, 1 PO BID). If on DSF, monolaurin 1 scoop/day and serrapeptase 2 capsules PO BID are substituted for Biocidin and Stevia. Triple probiotics (Theralac, 1 PO BID, Orthobiotic 1 PO BID; *saccharomyces boulardii*, 1 PO BID) NAC 600 mg PO BID; Alpha lipoic acid 600 mg PO BID; glutathione 1000 mg PO BID
Days 3 + 4 (pre-HDD CT)	Methylene blue 100 mg in the am Methylene blue 50 mg in the pm	Same as days 1 + 2
Days 5–8, HDD CT	Doxycycline 200 mg PO BID with meals (or Minocycline, up to 100 mg BID); hydroxychloroquine 200 mg PO BID with meals; rifampin 300 mg, 2 PO BID before meals; Nystatin 500,000 units 2 PO BID with meals; dapsone 100 mg, 2 PO BID with meals; methylene blue 100 mg PO BID with meals; leucovorin 25 mg, 3 PO BID with meals	Same as days 1 + 2, plus add L-methyl folate 15 mg, three PO BID, and increase glutathione to 2000 mg PO BID
Days 9–12 (post-HDD CT)	Stop all medication except methylene blue 100 mg PO BID and Leucovorin 25 mg, 3 PO BID	Same as days 5–8
Days 13–15 (post-HDD CT)	Decrease leucovorin to 2 PO BID and decrease methylene blue to 50 mg PO BID.	Same as days 5-8, except decrease L-methyl folate to 2 PO BID; decrease glutathione to 2000 mg PO QD.
Days 16–22 (week 3, post-HDD CT)	Stop methylene blue, decrease leucovorin to 25 mg PO BID	Probiotics are decreased to once a day; L-methyl folate is decreased to one PO BID; Glutathione is decreased to 500 mg twice a day (1000 mg total).
Week 4 (days 23 +)	Decrease leucovorin to 25 PO QD	Decrease L-methyl folate to one QD

We monitored patient symptoms at each follow-up consultation (q 1–2 months) including changes in eight major symptoms before and after HDDCT, side effects (clinical, laboratory) and subjective percentage of normal from baseline functioning. The number of pulsed HDDCT treatments varied based on response rates. Those who had significant clinical improvement without a relapse of underlying symptoms including fatigue, joint/muscle pain, headaches, neuropathy, insomnia, cognition, and sweating after one course of HDDCT remained off all antibiotic therapy. Those who demonstrated some clinical improvement but continued to have ongoing symptoms post-HDDCT, without any significant adverse side effects, did one to two more courses of HDDCT at least one month apart until reaching a plateau with no further improvement in eight major Lyme symptoms related to their CLD/PTLDS.

**Table 2 antibiotics-11-00912-t002:** Co-infection status and treatment response in 25 patients on HDDCT.

Response to Therapy	1 Course of HDDCT	2 Courses of HDDCT	3 Courses of HDD CT	LOI 1–4 yrs.	LOI 5–9 yrs.	LOI 10–20 yrs.	LOI 20+ yrs.	Age < 50	Age > 50	Bm	Bd	Bab FISH+	E	A	BartAB+	VEGF+	Bart PCR+	Bart FISH+	0 Co-inf	1 Co-inf	2 Co-inf	3 Co-inf	EM+	EM−
Remission	6	1	1	0	2	3	3	4	4	3	2	3	1	1	3	1	0	2	0	3	4	1	3	5
Improved 10–20%	6	1	2	0	3	2	4	6	3	2	2	4	4	0	6	2	1	2	0	2	3	4	3	6
Improved 21–30%	0	1	0	0	0	1	0	0	1	1	0	1	0	0	1	0	0	0	0	0	1	0	0	1
Improved >30%	2	0	1	0	0	3	0	1	2	0	1	1	0	1	2	0	0	0	1	0	1	1	1	2
No change	4	0	0	0	0	1	3	1	3	3	0	2	0	0	4	0	0	0	0	0	2	2	0	4

Abbreviations: Length of illness (LOI); Years (yrs.); *Babesia microti* (Bm); *Babesia duncani* (Bd); *Babesia* florescent in situ hybridization (Bab FISH); *Ehrlichia* (E); *Anaplasma* (A); *Bartonella* antibody (Bart AB); vascular endothelial growth factor (VEGF); *Bartonella* polymerase chain reaction (Bart PCR); *Bartonella* florescent in situ hybridization (Bart FISH); co-infections (Co-inf). *erythema migrans* (EM).

## Data Availability

All the data for this study are in the paper. There are no public archives containing information.

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
