# Peer review of "Efficacy of Short-Term High Dose Pulsed Dapsone Combination Therapy in the Treatment of Chronic Lyme Disease/Post-Treatment Lyme Disease Syndrome (PTLDS) and Associated Co-Infections: A Report of Three Cases and Literature Review"

_antibiotics, 2022, doi:10.3390/antibiotics11070912_

Round 1

Author Response

Reviewer 1:

  1. Much evidence stating Bartonella is carried by deer ticks, but not much evidence stating it is transmitted to humans from I. scapularis. Are there sources (not by the author) backing this up?

Answer: There is a European study in Ixodes Ricinus ticks regarding the transmission of one species of Bartonella, Bartonella birtlesii:

Reis C, Cote M, Le Rhun D, Lecuelle B, Levin ML, Vayssier-Taussat M, Bonnet SI. Vector competence of the tick Ixodes ricinus for transmission of Bartonella birtlesii. PLoS Negl Trop Dis. 2011;5(5):e1186. doi: 10.1371/journal.pntd.0001186. Epub 2011 May 31. PMID: 21655306; PMCID: PMC3104967.

“These results confirm the vector competence of I. ricinus for B. birtlesii and represent the first in vivo demonstration of a Bartonella sp. transmission by ticks. Consequently, bartonelloses should be now included in the differential diagnosis for patients exposed to tick bites.”

There are also studies showing transmission of 3 different Bartonella spp. in Ixodes Ricinus ticks with transovarial and transstadial transmission:

“The best engorgement results were obtained by ticks fed with B. henselae-spiked blood (65.3%) and B. schoenbuchensis (61.6%). Significantly more nymphs fed on infected blood (37.3%) molted into adults compared to the control group (11.4%). Bartonella DNA was found in 22% of eggs laid by previously infected females and in 8.6% of adults molted from infected nymphs. The transovarial and transstadial transmission of bartonellae suggest that I. ricinus could be a potential vector for three bacteria.

Król N, Militzer N, Stöbe E, Nijhof AM, Pfeffer M, Kempf VAJ, Obiegala A. Evaluating Transmission Paths for Three Different Bartonella spp. in Ixodes ricinus Ticks Using Artificial Feeding. Microorganisms. 2021 Apr 22;9(5):901. doi: 10.3390/microorganisms9050901. PMID: 33922378; PMCID: PMC8146832.

Regarding Ixodes ticks in the US: Bartonella spp. have been found in adult Ixodes pacificus ticks and may be an important reservoir:

“In these infected ticks, molecular analysis showed a variety of Bartonella strains, which were closely related to a cattle Bartonella strain and to several known human-pathogenic Bartonella species and subspecies: Bartonella henselae, B. quintana, B. washoensis, and B. vinsonii subsp. berkhoffii. These findings indicate that I. pacificus ticks may play an important role in Bartonella transmission among animals and humans.

               Chang CC, Chomel BB, Kasten RW, Romano V, Tietze N. Molecular evidence of Bartonella spp. in questing adult Ixodes pacificus ticks in California. J Clin Microbiol. 2001 Apr;39(4):1221-6. doi: 10.1128/JCM.39.4.1221-1226.2001. PMID: 11283031; PMCID: PMC87914.

Similarly, in this study: “Presence of Bartonella DNA was explored in 168 questing adult Ixodes pacificus ticks from Santa Cruz County, California. Bartonella henselae type I DNA was amplified from 11 ticks (6.55%); previously, two (1.19%) were found to be infected with Borrelia burgdorferi and five (2.98%) with Anaplasma phagocytophilum. Detection of B. henselae was not dependent on co-infection. The present study offers additional evidence that Ixodes spp. ticks may act as hosts and possibly vectors for B. henselae.”

Holden K, Boothby JT, Kasten RW, Chomel BB. Co-detection of Bartonella henselae, Borrelia burgdorferi, and Anaplasma phagocytophilum in Ixodes pacificus ticks from California, USA. Vector Borne Zoonotic Dis. 2006 Spring;6(1):99-102. doi: 10.1089/vbz.2006.6.99. PMID: 16584332.

Also, Bart spp have been found in Ixodes scapularis, but the vector competence has not yet been determined:

“Overall, 129/929 (13.9%) Ixodes ticks were PCR positive for Borrelia burgdorferi sensu stricto, 48/929 for B. bissettiae whereas 23/929 (2.5%) were PCR positive for a Bartonella henselae…PCR positive rates were highly variable depending on geographic location and tick species, with Ixodes affinis (n = 155) collected from North Carolina, being the tick species with the highest prevalence's for both Borrelia spp. (63.2%) and B. henselae (10.3%)

Maggi RG, Toliver M, Richardson T, Mather T, Breitschwerdt EB. Regional prevalences of Borrelia burgdorferi, Borrelia bissettiae, and Bartonella henselae in Ixodes affinis, Ixodes pacificus and Ixodes scapularis in the USA. Ticks Tick Borne Dis. 2019 Feb;10(2):360-364. doi: 10.1016/j.ttbdis.2018.11.015. Epub 2018 Nov 27. PMID: 30503356.

So the evidence to date has shown vector competence of Ixodes ticks to carry multiple bartonella species, transmit infection in Ixodes Ricinus ticks, and be suggestive of transmission in Ixodes scapularis and I. pacificus. I added a sentence and references in the text [1208-1210]. Thank you.

  1. Case 2.1: Was she acutely treated with doxy in 2012?

Answer: The physician treating her initially thought it was a spider bite, and chose not to treat her acutely, but when the Elisa returned positive with a 23 kda band on the W. blot, she was then given Penicillin.

  1. How long after treatment was this?

Answer: Her migratory pain in August 2012 was prior to treatment. The rash was in July 2012.

  1. What is 110% normal? Better than pre Lyme infection? Would that indicate another

issue other than Lyme being resolved?

Answer: Yes, she described her state of health as being the best she had ever felt in her life. Part of the improvement was probably due to her Babesia symptoms also being resolved, as she had no further night sweats. Babesia is known to increase underlying Lyme symptoms (Krause PJ, Telford SR 3rd, Spielman A, Sikand V, Ryan R, Christianson D, Burke G, Brassard P, Pollack R, Peck J, Persing DH. Concurrent Lyme disease and babesiosis. Evidence for increased severity and duration of illness. JAMA. 1996 Jun 5;275(21):1657-60. PMID: 8637139.).

So when the patient was seen August of 2021 and was 4 months symptom free post HDDCT, she stated her arthritis was gone, headaches were gone, there was no further fatigue with excellent energy, her sleep was great and night sweats were gone. At the follow up visit on 2/2022, that is when she stated “she is better than she has ever been”.

  1. Case 2.2: was tick-borne disease screening done?

Answer: No. The tick bite occurred at age 6 when he lived in Massachusetts, in a highly endemic area, and they were unable to get the head of the tick out. The patient went to a local ER where they removed it with a scalpel and wide excision, however neither the ER physician or local physician did a screen for Lyme and TBDs at that time. It wasn’t until around 2017 (the patient was 29 years old) when a psychiatrist seeing the patient ran tickborne testing and identified Lyme disease, Bartonella, testosterone deficiency, low immunoglob’s and other medical problems.

  1. Was this testing done @ 6 yrs old when he had multiple tick bites? Years later?

Answer: As above, it wasn’t until 23 years later when a psychiatrist ran tickborne testing.

  1. Case 2.3: But no changes in diet?

The patient remained on a low carbohydrate diet for his metabolic syndrome, and a low cholesterol diet for his hyperlipidemia. The diet remained the same, but once he stopped antibiotics, his cognitive symptoms significantly worsened.

  1. Reformatting of Table 1:

Response to

Therapy

1

Course of HDDCT

2 Courses of HDDCT

3 Courses of HDD

CT

LOI

1-4 yrs

LOI

5-9 yrs.

LOI 10-20 yrs.

LOI 20+ yrs. 

Age < 50

Age > 50

Bm

Bd

Bab FISH+

E

A

Bart

AB+

VEGF+

Bart

PCR+

Bart FISH+

0 Co-inf

1 Co-inf

2 Co-inf

3

Co-inf

EM+

EM-

Remission

6

1

1

0

2

3

3

4

4

3

2

3

1

1

3

1

0

2

0

3

4

1

3

5

Improved 10 – 20%

6

1

2

0

3

2

4

6

3

2

2

4

4

0

6

2

1

2

0

2

3

4

3

6

Improved 21 – 30%

0

1

0

0

0

1

0

0

1

1

0

1

0

0

1

0

0

0

0

0

1

0

0

1

Improved> 30%

2

0

1

0

0

3

0

1

2

0

1

1

0

1

2

0

0

0

1

0

1

1

1

2

No change

4

0

0

0

0

1

3

1

3

3

0

2

0

0

4

0

0

0

0

0

2

2

0

4

Reviewer 2 Report

Some repetitions of words in the same sentence could be improved accordingly. 

This manuscript entitled “Efficacy of Short-Term High Dose Pulsed Dapsone Combination Therapy in the Treatment of Chronic Lyme Disease/Post- 3 Treatment Lyme Disease Syndrome (PTLDS) and Associated 4 Co-infections: A Report of Three Cases and Literature Review” is a very good contribution towards the control of tick-borne diseases specially Lyme borreliosis. It is well organized providing a thorough understanding about the effect of the High Dose Pulsed Dapsone Combination in chronic Lyme disease therapy. In my opinion, the manuscript is suitable for the publication in this journal “antibiotics” after the authors have addressed the following comments:

Paper should be concise rather than providing so much detail as it seems hard to concentrate.

Methodology

1-Have the authors taken ethical approval? If yes then please mention in the methodology section.

2-why the authors didn’t use same dose of DSF?

3-why they used two courses in case 2 and why 3 in case 3? Why not same

4-Line 863-864           what could be the association between haemoglobin of the patient and HDDCT and what effect it would cause if the Hb level is not greater than 12?

5-Why methemoglobinemia occurs and how this protocol is affecting it?

6-Line 865-869 the authors have mentioned the side effects of dapsone but they didn’t explain whether these side effects were increased with the increase in dose level?

Results:

7-Line 947-949. Please rephrase the sentence “Ten patients (40%) had been ill for greater than 20 years, 10 patients (40%) had been ill between 10 to 20 years, 5 patients (20%) had been ill between 948 5 and 9 years, and no patients had been ill less than 5 years in duration”.

8-Have you observed any risk associated with the high dose of DSF?

Author Response

Reviewer 2:

Submission Date, 01 June 2022. Date of this review 14 Jun 2022 17:02:12

Comments and Suggestions for Authors

Some repetitions of words in the same sentence could be improved accordingly. 

This manuscript entitled “Efficacy of Short-Term High Dose Pulsed Dapsone Combination Therapy in the Treatment of Chronic Lyme Disease/Post-Treatment Lyme Disease Syndrome (PTLDS) and Associated Co-infections: A Report of Three Cases and Literature Review” is a very good contribution towards the control of tick-borne diseases specially Lyme borreliosis. It is well organized providing a thorough understanding about the effect of the High Dose Pulsed Dapsone Combination in chronic Lyme disease therapy. In my opinion, the manuscript is suitable for the publication in this journal “antibiotics” after the authors have addressed the following comments:

Paper should be concise rather than providing so much detail as it seems hard to concentrate.

Methodology

1-Have the authors taken ethical approval? If yes then please mention in the methodology section.

Answer: Yes. All patients on this protocol had signed informed consent forms that outlined the proposed benefits and potential risks of our study; patients volunteered to enroll in this high dose pulsed dapsone study at our medical center based on our prior research illustrating the benefit of dapsone combination therapy in the treatment of CLD/PTLDS [30] [18] [39], and on the drug’s documented action on “persister” bacteria in biofilms [34]. See lines 687-690

2-why the authors didn’t use same dose of DSF?

Answer: Patients have variable tolerance to DSF with some having severe Herxheimer reactions at low doses. Some patients can tolerate higher doses. We used a dose that was tolerated in each patient, which varied, but was individualized.

3-why they used two courses in case 2 and why 3 in case 3? Why not same

Answer: Case 2 had a flare up of neuropathy with DSF (line 414-415: As his neuropathy continued to flare up, even beginning with low dose disulfiram (62.5 mg/day), the DSF was stopped). So the use of DSF varied from patient to patient based on tolerance. As neuropathy from DSF is one of the worst potential side effects, we were careful to stop it in all patients with any signs of neuropathy. Case 3 was able to tolerate DSF better.

4-Line 863-864           what could be the association between haemoglobin of the patient and HDDCT and what effect it would cause if the Hb level is not greater than 12?

Answer: Hemoglobin levels tend to temporarily drop on dapsone because of the drugs effect on inhibiting folic acid metabolism. A 4-day pulse of HDDCT led to an average of less than a 2 gram drop of Hb (line 988-989: The risk of significant anemia during HDDCT was minimized using high dose folinic acid (75 mg PO BID) with high dose L-methylfolate (45 mg PO BID) and mean drops in hemoglobin of 1.75 grams during the 4-day pulse). We prefer higher hemoglobin levels when people start the protocol, to avoid significant anemia, but theoretically, someone could still take a 4 day pulse if the Hb was less than 12.

5-Why methemoglobinemia occurs and how this protocol is affecting it?

Answer: Lines 972-977: The four major side effects of HDDCT that are addressed in the dapsone protocol, using high dose folic acid (L-methyl-folate, Leucovorin), methylene blue, glutathione and antioxidants can be expressed as ‘Do no ‘H.A.R.M.’, i.e., Herxheimer reactions, Anemia, Rashes, and Methemoglobinemia. Herxheimer reactions are due to inflammatory cytokine release with spirochetal killing [50]; anemia, secondary to inhibition of folic acid metabolism, or hemolysis due to G-6-P-D deficiency [40]; rashes, if significant sulfa sensitivity exists, and methemoglobinemia, due to increased levels of oxidative stress with decreased oxygen-carrying capacity [86].

So dapsone routinely increases methemoglobin levels, but short-term use of high dose dapsone along with the use of methylene blue, glutathione and antioxidants helps to minimize the rise in methemoglobin, allowing the protocol to be tolerated and safe.

Lines 991-998: Administration of glutathione precursors (NAC 600 mg PO BID) and Alpha lipoic acid 600 mg PO BID with glutathione 2000 mg PO BID served to not only block NFKappaB and help decrease Herxheimer reactions and inflammatory cytokine production [88] [89] [56], but also support detoxification and anti-oxidant activity along with methylene blue. Methylene blue (MB) has several functions, primarily to help lower methemoglobin levels while on dapsone [86], apart from its effects on the biofilm/persister forms of Borrelia and Bartonella [90] [91]. Methemoglobin levels ranged between 0.3% to 23.9% (N = 17) while on methylene blue and glutathione, with a mean increase in methemoglobin of 7.97% during the 4-day period, and all methemoglobin levels returned to normal range (< 2%) when checked three to four weeks off treatment.

6-Line 865-869 the authors have mentioned the side effects of dapsone but they didn’t explain whether these side effects were increased with the increase in dose level?

Answer: Yes, an increase in dapsone would normally increase anemia, Herxheimer reactions and methemoglobin levels, which are the major side effects of dapsone, but we didn’t routinely use methylene blue in all patients in our initial paper using double dose dapsone.  In our prior double dose dapsone combination therapy paper, it says: “in other chronically ill Lyme-MSIDS patients given dapsone at 100 mg or higher [14], oral methylene blue was occasionally needed, and effective in keeping methemoglobin levels below 5%, allowing continuation of therapy.”

So we added methylene blue routinely when using high dose dapsone therapy, and mean MetHb levels were 7.97%. We increased folic acid supplementation in HDDCT, compared to DDCT to help decrease potential anemia, and also increased glutathione to address Herxheimer reactions and potential elevations in MetHb. I added a sentence to reflect that higher dosing of folic acid, methylene blue and glutathione were used in those patients on HDDCT compared to DDDCT, as higher doses of dapsone would potentially increase Herxes, anemia and methemoglobinemia.

Lines 998-1000 now says: Higher dosing of folic acid, methylene blue and glutathione were used in those patients on HDDCT compared to DDDCT [39], to help prevent an increase in Herxheimer reactions, anemia and methemoglobinemia, with favorable outcomes.

Thank you for pointing that out. It’s an important distinction.

Results:

7-Line 947-949. Please rephrase the sentence “Ten patients (40%) had been ill for greater than 20 years, 10 patients (40%) had been ill between 10 to 20 years, 5 patients (20%) had been ill between 948 5 and 9 years, and no patients had been ill less than 5 years in duration”.

Answer: line 764-766, now reads: Forty % of patients (N=10) had been ill for greater than 20 years, 40% (N=10) had been ill between 10 to 20 years, 20% (N=5) had been ill between 5 to 9 years, and no patients had been ill for less than 5 years in duration.

8-Have you observed any risk associated with the high dose of DSF?

Answer: I have seen temporary increases in fatigue, brain fog, joint and muscle pain and neuropathy, with Herxheimer reactions, but fortunately, I have not seen any long term complications. The risk increases with DSF dose, and we kept the dose down. Lines 721-728: “Several patients took disulfiram (DSF) in combination with HDDCT, if they had previously failed to have an adequate clinical improvement with either drug regimen used alone or in combination. Patients signed a consent informing them of the potential benefits and risks of DSF [32] [33], including increased fatigue, brain fog/cognitive dysfunction, worsening psychiatric symptoms, liver function abnormalities, and/or increased neuropathy [49]. They were instructed to stop DSF immediately if there was any worsening in underlying neuropathic symptoms, and to contact our office immediately if any severe adverse effects were noted. The dose of DSF used in prior clinical studies ranged from lower dose DSF (250 mg or less) to higher dose DSF (500 mg/day) [32] [33]. The majority of patients who took DSF in our study used doses of 250 mg/day or less, to minimize potential adverse effects and Herxheimer reactions.

Finally, I added a new study that was just published on the global seroprevalence of Bb sensu lato at 14.5% at the very end of the article. Please see lines 1223-1224.

Thank you, and please let me know if you have any other questions or suggestions.

Dr Richard Horowitz
